# scm⁶A-seq reveals single-cell landscapes of the dynamic m⁶A during oocyte maturation and early embryonic development

Huan Yao [1,2,8], Chun-Chun Gao [1,8], Danru Zhang[3,8], Jiawei Xu[4,8], Gege Song[1], Xiu Fan[1], Dao-Bo Liang[1], Yu-Sheng Chen [1], Qian Li[2], Yanjie Guo[2], Yu-Ting Cai[1,5], Lulu Hu[6], Yong-Liang Zhao [1,5], Ying-Pu Sun [2] ✉, Ying Yang [1,5,7] ✉, Jianyong Han [3] ✉ & Yun-Gui Yang [1,5,7] ✉

N⁶-methyladenosine (m⁶A) has been demonstrated to regulate RNA metabolism and various biological processes, including gametogenesis and embryogenesis. However, the landscape and function of m⁶A at single cell resolution have not been extensively studied in mammalian oocytes or during preimplantation. In this study, we developed a single-cell m⁶A sequencing (scm⁶A-seq) method to simultaneously profile the m⁶A methylome and transcriptome in single oocytes/blastomeres of cleavage-stage embryos. We found that m⁶A deficiency leads to aberrant RNA clearance and consequent low quality of *Mettl3^Gdf9* conditional knockout (cKO) oocytes. We further revealed that m⁶A regulates the translation and stability of modified RNAs in metaphase II (MII) oocytes and during oocyte-to-embryo transition, respectively. Moreover, we observed m⁶A-dependent asymmetries in the epi-transcriptome between the blastomeres of two-cell embryo. scm⁶A-seq thus allows in-depth investigation into m⁶A characteristics and functions, and the findings provide invaluable single-cell resolution resources for delineating the underlying mechanism for gametogenesis and early embryonic development.

N⁶-methyladenosine (m⁶A) is one of the most important RNA modifications for the eukaryotic transcriptome[1]. m⁶A regulates mRNA alternative splicing[2], export[3], genome accessibility[4], and transposon activity[5,6] in nucleus, and also affects mRNA translation efficiency, degradation, and stability[7] in cytoplasm. m⁶A is present along the whole transcriptome, and mainly enriched in coding region and 3' UTR on the mRNAs[8–12]. Thus, m⁶A has been widely studied to exert extensive regulatory functions in gametogenesis and embryonic developmental events[13].

Many studies have highlighted the roles played by m⁶A during oocyte development. In zebrafish, the m⁶A modified maternal transcripts tend to be recognized by *Ythdf2* and degraded faster than the unmethylated transcripts[14]. In mice, *Mettl3* deficiency led to a highly sterile phenotype in females, with features of abnormal oocyte

¹CAS Key Laboratory of Genomic and Precision Medicine, Collaborative Innovation Center of Genetics and Development, College of Future Technology, Beijing Institute of Genomics, Chinese Academy of Sciences and China National Center for Bioinformation, Beijing 100101, China. ²Center for Reproductive Medicine, Henan Key Laboratory of Reproduction and Genetics, The First Affiliated Hospital of Zhengzhou University, Zhengzhou, China. ³State Key Laboratory for Agrobiotechnology, College of Biological Sciences, China Agricultural University, Beijing 100193, China. ⁴The First Affiliated Hospital of Zhengzhou University, Zhengzhou, China. ⁵Sino-Danish College, University of Chinese Academy of Sciences, Beijing 101408, China. ⁶Fudan University Institutes of Biomedical Sciences, Shanghai Cancer Center, Shanghai Key Laboratory of Medical Epigenetics, International Co-laboratory of Medical Epigenetics and Metabolism (Ministry of Science and Technology), Shanghai Medical College of Fudan University, Shanghai 200032, China. ⁷Institute of Stem Cell and Regeneration, Chinese Academy of Sciences, Beijing 100101, China. ⁸These authors contributed equally: Huan Yao, Chun-Chun Gao, Danru Zhang, Jiawei Xu. ✉e-mail: syp2008@vip.sina.com; yingyang@big.ac.cn; hanjy@cau.edu.cn; ygyang@big.ac.cn

ovulation and disordered oocyte maturation[15]. *Ythdf2* depletion resulted in female-specific infertility and maternal transcript dosage during meiotic maturation[16]. The m⁶A methyltransferase complex component KIAA1429 is highly expressed in germinal vesicle (GV) oocytes and *Kiaa1429* depletion in female mice showed small ovaries, accompanied by significantly decreased numbers of surrounded nucleolus (SN) oocytes and abnormal oocytes competence[17]. Moreover, in the cleavage-stage embryos, maternal *Mettl3* depleted zygotes were arrested at the 1-cell stage[15]. Zygotic *Mettl3* knockdown led to the failure of zygotic genome activation (ZGA), and the decreased m⁶A abundance from the GV oocytes to the 2-cell embryos[18].

ZGA of the mice zygote embryos begins at the late 1-cell embryos, and many studies have pointed to the significance of chromatin remodeling in zygotic transcription initiation, including DNA methylation, histone modifications, chromatin accessibility, high-order chromatin structures formation, and RNA polymerase II (Pol II) binding[19-22]. Interestingly, in addition to the newly synthesized zygotic transcripts, ZGA-dependent maternal RNA clearance processes are crucial for early embryonic development[23,24]. Moreover, RNA metabolism during ZGA results in the heterogeneity of epigenomic modification among blastomeres and suggests transcriptome-wide regulation initiated at the 2-cell embryonic stage[25,26].

Previous studies have depicted the landscapes of the single-cell transcriptome[27-29], and low input-based translatome[30,31] and m⁶A methylome[32] in oocyte and early stages of embryos. But the characteristics and functions of RNA methylation at single-cell level remain unclear. Accumulated methods, which are mainly based on antibody enrichment, sensitive enzymatic digestion or chemical conversion, have been developed to profile and characterize transcriptome-wide m⁶A from bulk to low-input samples[8,32-37]. Recently, scDART-seq[38] using exogenously expressed APOBEC1-YTH construct showed the power of single-cell m⁶A detection, opening a new avenue for investigating epitranscriptomic regulation at the single-cell level.

In this study, we developed a single-cell m⁶A sequencing (scm⁶A-seq) method without exogenous gene expression. The scm⁶A-seq combines the principle of RNA multiplex labeling technology[39] with methylated RNA immunoprecipitation sequencing (MeRIP-seq/m⁶A-IP), enabling the capture of the transcriptome-wide m⁶A landscape and comparison of the m⁶A level among single cells. Utilizing scm⁶A-seq, we interrogate the m⁶A modification of mouse oocytes and early embryos. Our findings revealed dynamic m⁶A regulation in RNA metabolism at single-cell resolution during oocyte development and ZGA.

## Results

### Establishment of the single-cell m⁶A sequencing method
To profile the whole RNA m⁶A modified transcriptome in single cells, we developed the single-cell m⁶A sequencing (scm⁶A-seq) method, in which the RNAs from each cell were first fragmented and ligated, and then labeled with two rounds of barcoded DNA adapters for parallelized single-cell sequencing. The first round of labeling involved ligating the adapters with barcode 1 to the fragmented RNAs from each cell, followed by the second round of labeling to add the barcode 2 adapters to the 3' end of barcode 1 adapters through base-paring (Supplementary Fig. 1a). Then, the barcoded RNAs from different cells were pooled together and subjected to RNA-seq and m⁶A immunoprecipitation (m⁶A-IP), respectively ("Methods"). During library construction, a dSpacer-blocked template switch oligo (TSO) and T7 primer were used for cDNA synthesis and amplification, and then, T7 was used for in vitro transcription, ribosomal RNA (rRNA) depletion, and PCR amplification and sequencing (Fig. 1a and Supplementary Fig. 1a). Overall, scm⁶A-seq enables simultaneous transcriptome and m⁶A methylome sequencing at single-cell resolution through specific barcode labeling.

To determine whether scm⁶A-seq can be used to identify m⁶A peaks in single cells, we first performed scm⁶A-seq on germinal vesicle (GV) oocytes. Overall, we obtained an average of 12 million mapped

reads and more than 15,000 detected RNAs in each of 10 GV oocytes (Fig. 1b). High expression correlation was observed among 10 individual GV oocytes, especially in the RNA-seq data (Supplementary Fig. 1b), justifying the reproducibility and sensitivity of scm⁶A-seq. Attributing to the sufficient sequencing depth and coverage, we further identified high-confident m⁶A features ("Methods"), and obtained more than 5000 m⁶A modified peaks in over 4000 RNAs in each oocyte (Fig. 1c and Supplementary Data 1). To validate the accuracy and specificity of the detected m⁶A peaks by scm⁶A-seq, we also performed RNA-seq and methylated RNA immunoprecipitation (MeRIP-seq) using tens of oocytes (25, 50, and 70) and used the same criteria for filtering the peaks (Supplementary Data 2). More than 80% of m⁶A modified RNAs obtained by both single-cell and bulk sequencing methods were protein-coding genes, and a slightly higher proportion in scm⁶A-seq (Supplementary Fig. 1c), and the modified peaks were mainly located in their coding sequence (CDS), stop codon regions and 3'UTRs (Fig. 1d). Metagene analysis of the identified m⁶A peaks in oocytes showed an obvious enrichment of m⁶A signals around transcription start sites (TSS) and stop codon regions (Fig. 1e), and conserved RRAC motif (R = A/G) was observed in high-confidence m⁶A peaks (Supplementary Fig. 1d), which are consistent with previous reports[15].

We further characterized the m⁶A features in single cell identified by scm⁶A-seq. First, we detected a good repeatability of m⁶A modified RNAs identified from tens of GV oocytes using the same method of scm⁶A-seq (Supplementary Fig. 1e). We then counted the m⁶A modified RNAs among 10 individual GV oocytes and found that over 40% m⁶A modified RNAs can be identified in more than 5 oocytes (Fig. 1f). Moreover, we also observed that the expression level of m⁶A modified RNAs was increased along with their frequencies in 10 individual oocytes (Fig. 1g). Comparing the modified RNAs in single cell with those in tens of oocytes sample, we found that the conserved modified RNAs with high frequencies was more likely to be detected in 70 GVs samples, such as 1 out of 10 oocytes in scm⁶A-seq with 40.96%, 4/10 with 78.46%, 6/10 with 87.94%, and 10/10 with 97.60% (Fig. 1h). Furthermore, we defined the commonly modified RNAs as those with m⁶A peak enrichment scores in the top 5000 for each oocyte and meanwhile, identified in no less than 6 oocytes. A total of 1816 commonly modified RNAs were categorized into this group accounting for more than 60% of m⁶A modified RNAs in each oocyte, in relative to only around 40% in tens of oocytes sample (Supplementary Fig. 1f). Importantly, nearly 70% of the modified RNAs identified by bulk MeRIP-seq[15] and 55% by low input sequencing (ULI-MeRIP-seq)[32] of GV oocytes were also detected in merged single GV oocyte by scm⁶A-seq (Supplementary Fig. 1 g). These commonly modified RNAs identified by three detection techniques were mainly involved in cell cycle, DNA repair, and cell division pathways (Supplementary Fig. 1h), such as *E2f1* and *Bod1* (Supplementary Fig. 1i). Notably, some modified RNAs were identified in most proportion of cells only by scm⁶A-seq, but not by low input and bulk sequencing, such as the genes of *Hist4h4*, *Hist1h4f*, and *Mapk3* involving cell cycle and DNA replication (Supplementary Fig. 1i). Thus, scm⁶A-seq enables robust m⁶A identification in single cells and is more sensitive especially for low-abundant modified RNAs.

### METTL3 promotes degradation of methylated RNAs in GV oocytes
In *Mettl3^Gdf9* conditional knockout (cKO) mice (*Mettl3^flox/flox*;*Gdf9-cre*), the ovary size was largely decreased and meanwhile the oocyte maturation was also blocked. Over 80% of the oocytes were arrested at GVBD stage and the percentage of MII oocytes was less than 5%. Moreover, zygotes derived from *Mettl3^Gdf9* cKO oocytes were arrested at 1-cell stage after fertilization[15]. But whether this phenotype is mediated by METTL3-dependent m⁶A in oocytes remained to be determined. The scm⁶A-seq was then performed on control and *Mettl3^Gdf9* cKO oocytes in the GV stage to identify METTL3-dependent modified RNAs. We observed globally decreased m⁶A signals in

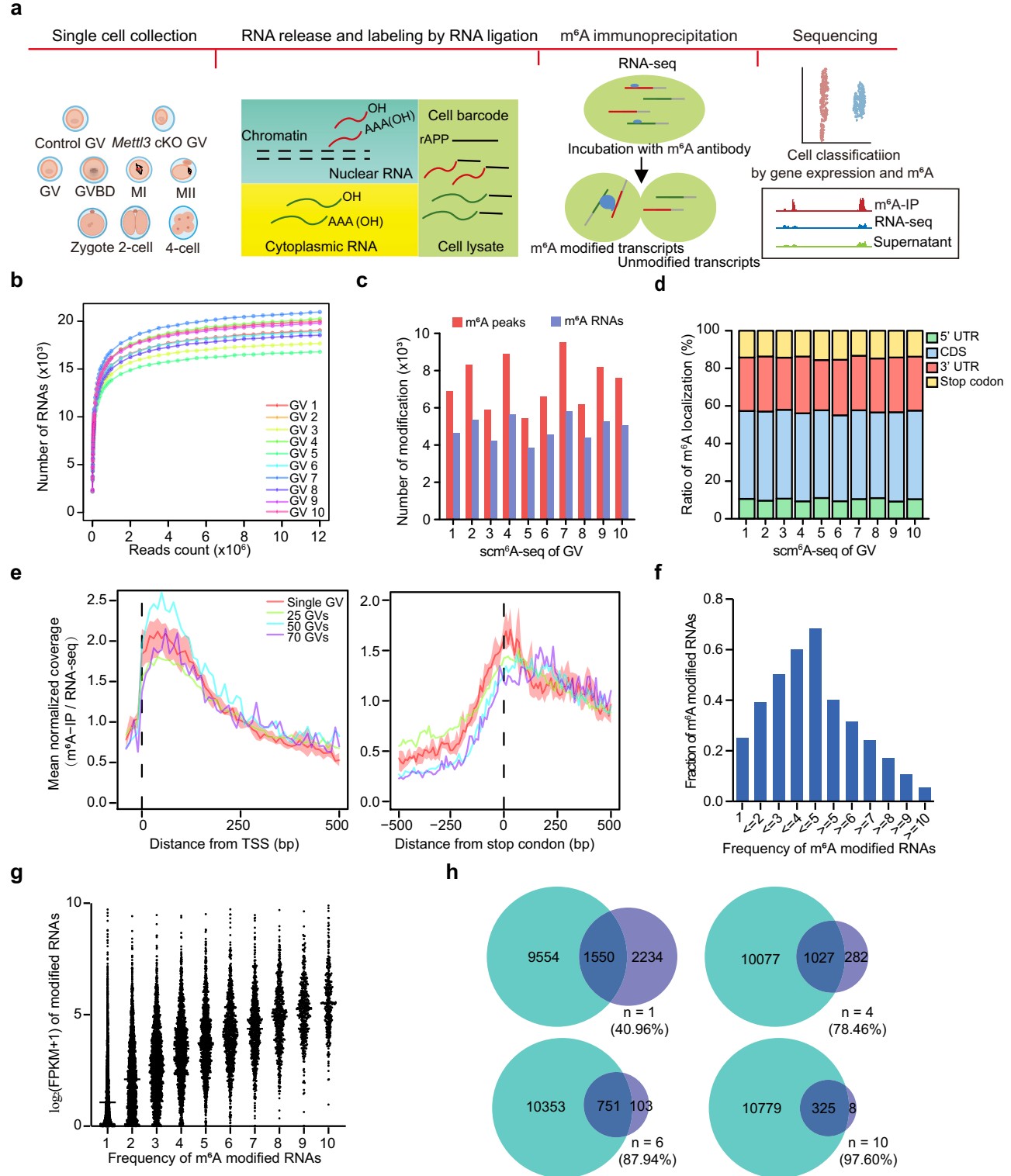

**Fig. 1 | scm⁶A-seq detects m⁶A signatures in individual germinal vesicle (GV) oocytes. a** Schematic diagram showing the scm⁶A-seq method and an overview of the samples analyzed. The carton image of oocytes and early embryos is created with BioRender. **b** Saturation curves for each GV oocyte data obtained by scm⁶A-seq. Each point on the curve was generated by randomly selecting a number of raw reads from each GV cell library and then using the same alignment pipeline to call covered genes. **c** Bar plot showing the number of detected m⁶A peaks and modified transcripts in each GV oocyte using scm⁶A-seq. **d** The distribution of mRNA m⁶A peaks in individual GV oocytes. **e** Metagene profiles depicting the normalized sequence coverage ratio of m⁶A-IP to RNA-seq data in positions surrounding the transcription start site (TSS) (left) and translation stop codon (right). The red line and light red shadow represent the mean and quantiles of 10 GV oocytes determined by

scm⁶A-seq, respectively. The other three colored lines represent bulk MeRIP-seq and RNA-seq, as labeled in the panel. **f** Bar plot displaying the accumulative fraction of the number of the commonly m⁶A modified transcripts in 10 GV oocytes. Source data are provided as a Source data file. **g** The box and point plot showing the expression of common m⁶A modified transcripts. The x-axis shows the number of common m⁶A peaks in 10 GV oocytes. Source data are provided as a Source data file. **h** Venn diagram displaying the common m⁶A modified transcripts between single cells and bulk cells detected from MeRIP-seq data. Light green pie represents identified modified RNAs identified in 70 GV oocytes by MeRIP-seq, and the purple pies represent the common m⁶A modified RNAs form 10 individual GV oocytes as determined by scm⁶A-seq. n represents the frequency with which common m⁶A modified RNAs appeared in the analysis of the 10 individual GV oocytes.

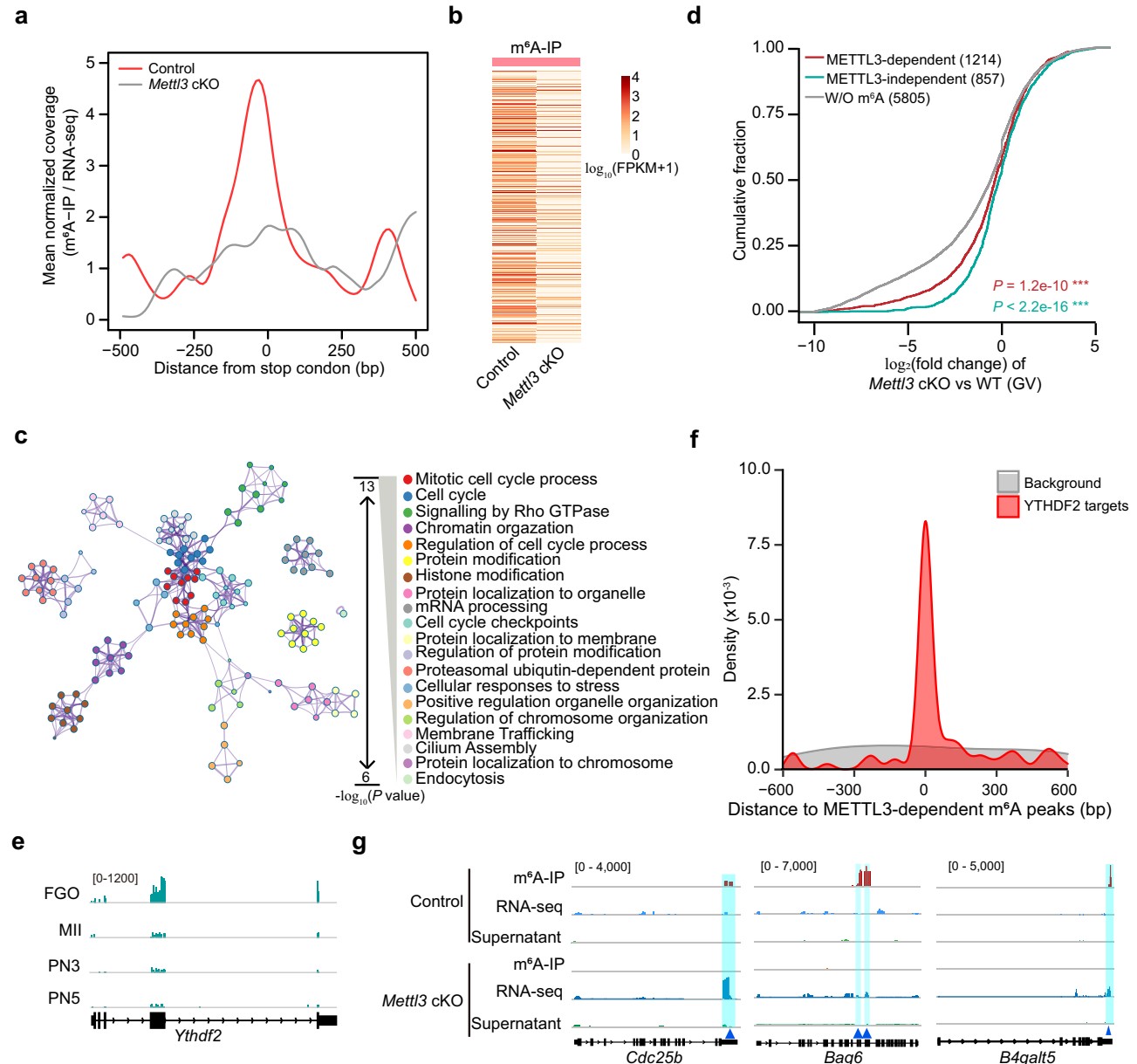

**Fig. 2 | METTL3-catalyzed m⁶A deposition mediates the degradation of m⁶A modified RNAs in GV oocytes. a** Metagene profiles depicting m⁶A signals surrounding the stop codon in control and *Mettl3* conditional knockout (cKO) GV oocytes. **b** Heatmap showing the METTL3-dependent m⁶A modified transcripts in GV oocytes. **c** Functional annotation analysis of METTL3-dependent m⁶A modified transcripts using Metascape. The enrichment and *P* value was calculated with default parameters using hypergeometric test of functional annotation in DAVID database. **d** Cumulative frequency of expression change (log₂(fold change)) of METTL3-dependent, METTL3-independent m⁶A modified and unmodified RNAs upon *Mettl3* silencing. *P* values were determined by one-sided Wilcoxon test.

$P = 1.2e–10$, between METTL3-independent and W/O m⁶A RNA sets. $P < 2.2e–16$, between METTL3-dependent and W/O m⁶A RNA sets. ***$P < 0.001$. **e** Integrated Genomics Viewer (IGV) diagram displaying the translation signals of *Ythdf2* in oocytes and zygotes. **f** Density plot displaying the distance between YTHDF2 target peaks detected by enhanced crosslinking and immunoprecipitation sequencing (eCLIP-seq) in mESCs[42] (GSE151788) and METTL3-dependent m⁶A peaks in GV oocytes by scm⁶A-seq. **g** Integrated Genomics Viewer (IGV) diagram showing the METTL3-dependent m⁶A peaks across *Cdc25b*, *Bag6*, and *B4galt5*. The light blue boxes represent the identified m⁶A peaks identified by scm⁶A-seq.

*Mettl3^Gdf9* cKO oocytes (Supplementary Fig. 2a), especially near stop codons (Fig. 2a). A total of 1214 RNAs, including over 5% modified ncRNAs (Supplementary Fig. 2b), had lost m⁶A signatures in *Mettl3^Gdf9* cKO GV oocytes, which were defined as the METTL3-dependent m⁶A modified RNAs (Fig. 2b, Supplementary Data 3). These RNAs in control oocytes showed m⁶A peaks embedded in conserved RRACH motif sequences (R = A/G, H = A/C/G) (Supplementary Fig. 2c) and significantly enriched in mitotic cell cycle and cell division processes (Fig. 2c), indicating that the cell cycle related modified RNAs are probably implicated in oocytes maturation. To further investigate the

role of m⁶A in regulating RNA metabolism in oocytes, we compared the expression changes of METTL3-dependent m⁶A-modified RNAs with the unmodified or modified ones independent of METTL3 in *Mettl3^Gdf9* cKO versus the control oocytes. The results showed that the expression of METTL3-dependent m⁶A-modified RNAs in *Mettl3^Gdf9* cKO oocytes is significantly enhanced in relative to the unmodified RNAs (Fig. 2d), suggesting that METTL3-dependent m⁶A-modified RNAs are preferentially degraded in control GV oocytes, but stabilized in *Mettl3^Gdf9* cKO oocytes owing to the loss of m⁶A modification. Intriguingly, significantly increased expression of METTL3-independent

m[6]A-modified RNAs relative to the unmodified RNAs was also observed (Fig. 2d). The RNA abundance of m[6]A modified RNAs is higher than the unmodified ones (Supplementary Fig. 2d), which is consistent with the previous report in single cell[38] and may be the reason for the high expression of METTL3-independent m[6]A-modified RNAs in *Mettl3* cKO oocytes. Since YTHDF2 has been identified as a m[6]A reader protein mediating the degradation of its targeted m[6]A modified RNAs[16,40,41], and substantially higher translation signal[30] was observed in the fully-grown oocytes (Fig. 2e), we further analyzed the YTHDF2 binding sites from the public YTHDF2 enhanced crosslinking and immunoprecipitation sequencing (eCLIP-seq) data previously generated from mouse embryonic stem cells (mESCs)[42] and found that METTL3-dependent m[6]A signatures significantly overlap with the YTHDF2 targets compared with the random background (Fig. 2f). Furthermore, we identified significantly upregulated RNA expression upon *Mettl3[Gdf9]* cKO within the METTL3-dependent m[6]A modified RNA populations (Supplementary Fig. 2e), and found that these RNAs, such as *Cdc25b*, *Bag6*, *B4galt5*, *Strip1*, and *Smn1*, were involved in cell cycle and metabolism processes (Fig. 2g and Supplementary Fig. 2f). Collectively, our results suggested that METTL3-catalyzed m[6]A mainly regulates RNA stability and preferentially promotes degradation in GV oocytes.

## scm[6]A-seq distinguishes SN and NSN GV oocytes

The successful identification of m[6]A modification in single cells by scm[6]A-seq allows us to further classify the oocytes populations (30 control and 27 *Mettl3[Gdf9]* cKO GV oocytes) using m[6]A signatures. Through scm[6]A-seq, we not only barcoded individual cells prior to m[6]A enrichment, but also pooled the barcoded cells together in the same batch for library construction and sequencing, which allowed direct comparison of the m[6]A levels among individual cells using the data obtained from the same batch. In addition, scm[6]A-seq enables joint profiling of m[6]A methylome and transcriptome in individual cells, producing RNA-seq, m[6]A-IP and supernatant data for each cell (Supplementary Fig. 1a). We found that the control GV and *Mettl3[Gdf9]* cKO GV oocytes can be distinguished by using each expression matrix of RNA-seq, m[6]A-IP and supernatant or their merged expression matrix from scm[6]A-seq data in single cells (Fig. 3a–c and Supplementary Fig. 3a). Meanwhile, further including the single-cell m[6]A level in the matrix improved the accuracy in distinguishing the control and *Mettl3[Gdf9]* cKO GV oocytes due to a decreased m[6]A level in the *Mettl3[Gdf9]* cKO oocytes (Fig. 3d). Intriguingly, the cells were clustered into three classes when only transcriptome data were used (Supplementary Fig. 3b), but we observed two sub-populations in the control and *Mettl3[Gdf9]* cKO oocyte groups when taking the m[6]A level of individual oocytes as a third component for clustering (Fig. 3e and Supplementary Fig. 3c). Previous study reported that there are both SN and NSN GV oocytes on the basis of their growth-related transcription in the ovary of the mice after ovulation induction, with the NSN oocytes displaying lower developmental capacity[43]. Thus, the sub-populations in control and *Mettl3[Gdf9]* cKO groups were considered to correspond to SN and NSN oocytes which were further validated by the marker genes of *Gata3* and *Mapk7* for oocyte growth[44], respectively (Fig. 3f). Meanwhile, we also observed a lower RNA abundance in NSN oocytes, corresponding to their transcription activity and incomplete growth (Fig. 3g). We also found a higher proportion of NSN oocytes in *Mettl3[Gdf9]* cKO group (Supplementary Fig. 3d). And we also verified the existence of SN and NSN oocytes in the cumulus cell-oocyte complex (COC) by nucleus staining of both control and *Mettl3[Gdf9]* cKO mice (Fig. 3h). Thus, the multi-omics data from scm[6]A-seq enables more precise classification of cell populations. To further clarify the differences between SN and NSN oocytes, we analyzed the most significantly differentially expressed RNAs between SN and NSN oocytes, and found that most of these RNAs were related to oocyte growth (Fig. 3i). On the other hand, the dynamic expression of m[6]A methyltransferases and readers during

oocyte growth also suggested a m[6]A related mechanism of RNA metabolism during oocyte growth[45] (Supplementary Fig. 3e). As tri-methylated histone 3 lysine 36 (H3K36me3) has been reported to be involved in regulating gene expression during oocyte development[46], we defined the decreased genes (log₂fragments per kilobase of transcript per million mapped reads (FPKM) (fully-grown oocyte (FGO)/postnatal day 7 (P7)) < −2)) which lost the H3K36me3 during oocyte growth as oocyte development-silencing genes (Supplementary Fig. 3f). Because of the m[6]A-mediated RNAs degradation in GV oocytes (Fig. 2), we then evaluated the stability of oocyte development-silencing RNAs by measuring altered expression in control and *Mettl3[Gdf9]* cKO SN GV oocytes and observed abnormal accumulation of development-silencing RNAs in *Mettl3[Gdf9]* cKO oocytes (Supplementary Fig. 3g, h). These results suggest that the failure of degradation of development-silencing RNAs might be the crucial reason for the low quality of *Mettl3[Gdf9]* cKO GV oocytes.

## The dynamic m[6]A methylome during oocyte maturation

The mammalian oocyte maturation process is coupled with maternal RNA translational activation and degradation[47,48], including GV, germinal vesicle breakdown (GVBD) after the resumption of oocyte meiosis, and two consecutive metaphase (MI and MII) stages. Fully-grown GV oocytes are transcriptionally silent, making the oocytes maturation an ideal model for post-transcriptional regulation detection. To determine the regulation of m[6]A deposition during oocyte maturation, we performed scm[6]A-seq on cultured oocytes in the GV, GVBD, MI, and MII stages (10 oocytes for each stage) (Supplementary Data 4). We first summarized the changed RNAs expression during oocyte maturation using K-means clustering from RNA-seq of scm[6]A-seq, and obtained a cluster of degraded RNAs related to the cell cycle and oocyte maturation (Fig. 4a, b) and another set of RNAs with increased expression correlated with translation and protein transport (Supplementary Fig. 4a, b), which findings are consistent with the tendency of RNA metabolism during oocyte maturation. However, RNA expression matrix alone cannot accurately distinguish the stage-specific cells during oocyte maturation (Supplementary Fig. 4c). Therefore, we next depicted the m[6]A signatures of oocyte in different maturation stages by scm[6]A-seq (Supplementary Fig. 4d) and observed some stable m[6]A modified RNAs during oocytes maturation, such as *Lars2* (Supplementary Fig. 4e), involving in the signaling pathways (Supplementary Fig. 4f). Additionally, we also observed many stage-specific m[6]A-modified RNAs, such as *Bnc2*, *Tmed9*, *Arcn1*, and *Cd320* (Supplementary Fig. 4g), which were involved in differential functions, especially nucleosome assembly, transcription in GV oocytes, and spindle assembly, protein transport in M-phases oocytes (Supplementary Fig. 4h). These results suggest a non-co-transcriptionally regulatory mechanism of m[6]A during oocyte maturation. Interestingly, we observed an increased m[6]A enrichment during the oocyte maturation based on the metagene analysis and estimation of m[6]A level (Fig. 4c and Supplementary Fig. 4i). What's more, oocytes of different stages were well clustered when m[6]A features were included (Fig. 4d). Because oocyte maturation is accompanied by maternal RNA translational activation, we assessed the translation differences and found the m[6]A modified transcripts tend to have significantly higher translation levels than the unmethylated transcripts (Fig. 4e). We then aimed to determine if this process is regulated by the m[6]A readers of YTHDF1 and YTHDF3, which were validated with capability of enhancing the translation of m[6]A modified transcripts[49,50]. Based on public proteomic data generated from oocyte maturation[51], we observed a high level of YTHDF3 versus a low level of YTHDF1 in MII stage (Fig. 4f). In addition, we also observed considerate overlap between the m[6]A peaks and the YTHDF3 targets by eCLIP-seq of mESCs[42] (Fig. 4g). Collectively, these results indicate that m[6]A promotes the translation of m[6]A modified transcripts in MII stage, and this effect is potentially mediated by YTHDF3.

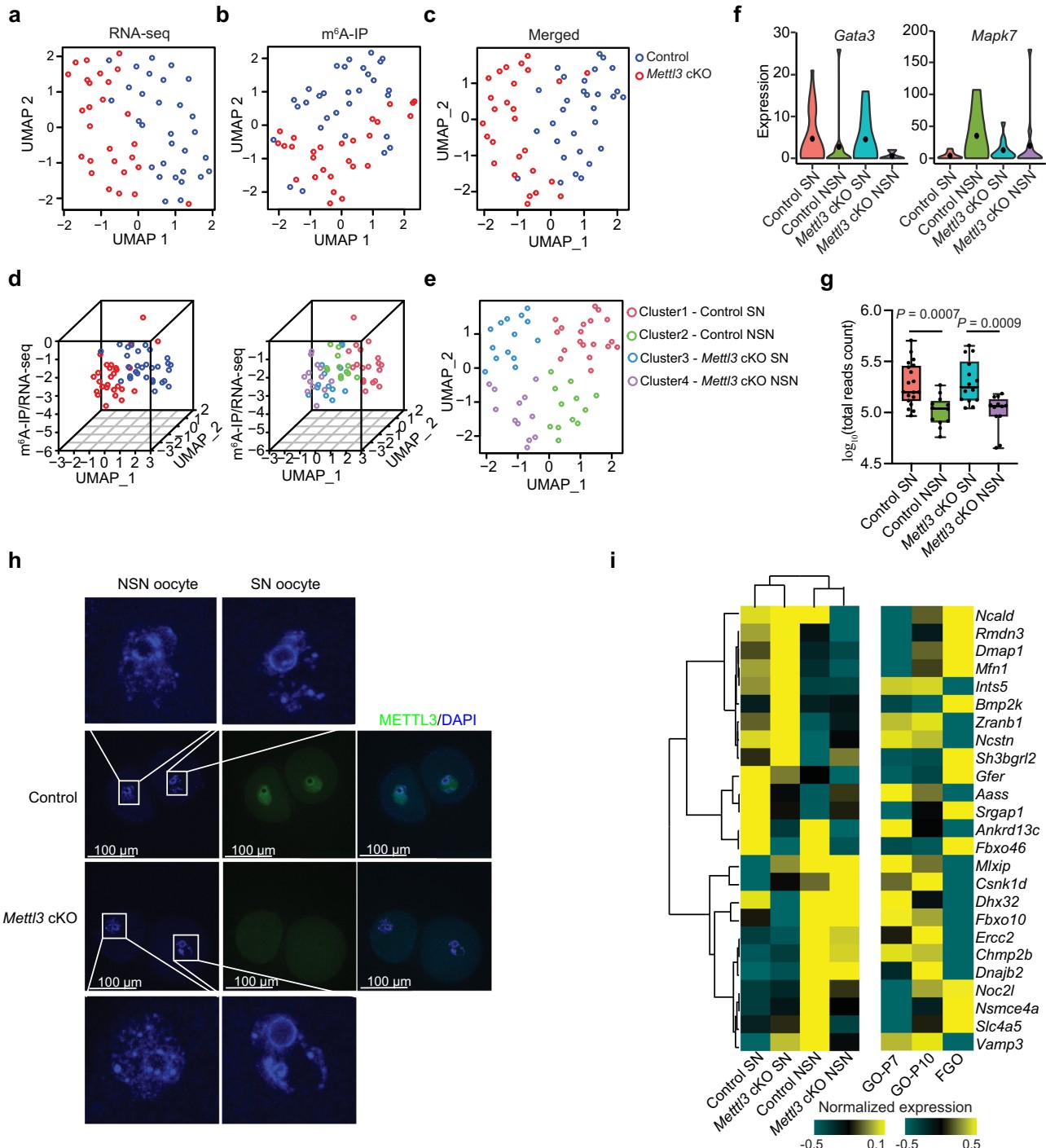

**Fig. 3 | scm⁶A-seq distinguishes surrounded nucleolus (SN) and non-surrounded nucleolus (NSN) oocytes. a–c** Uniform manifold approximation and projection (UMAP) plot based on the reads count matrix of RNA-seq (**a**) and m⁶A-IP (**b**), and merged total reads count of single GV oocytes (**c**). The UMAP plot is colored on the basis of the cells type. Source data are provided as a Source data file. **d** The 3D plot showing the relative m⁶A level of each GV oocyte based on the UMAP results, colored by cells type (left) and cells reclassified by the comprehensive analysis of the gene expression matrix reduced dimensions by UMAP and m⁶A level (right). Source data are provided as a Source data file. **e** The two-dimensional UMAP diagram showing the cell populations on the basis of the classification results in (**d**). The Clusters 1–4 of cells were redefined as Control SN, Control NSN, *Mettl3* cKO SN, and *Mettl3* cKO NSN clusters according to the expression of marker RNAs and genome type of oocytes. Source data are provided as a Source data file. **f** Violin plot showing the quality of markers *Gata3* and *Mpak7* expression in oocytes in different clustering populations. **g** Combined box and scatter plot showing the difference in RNA

amount between SN and NSN oocytes. *n* (Control SN) = 19, *n* (Control NSN) = 12, *n* (*Mettl3* cKO SN) = 14, and n (*Mettl3* cKO) = 12. The middle lines of the boxes represent the medians of datasets. The upper and bottom lines of the boxes are respectively the upper quantile and the lower quantile of the data. The whiskers mark the upper and lower limits of these datasets, respectively. The two-sided *P* value was calculated by unpaired student *t* test. Source data are provided as a Source data file. **h** Morphology of the pronuclear in both the SN and NSN oocytes isolated from the cumulus cell-oocyte complex (COC). Both control and METTL3-null oocytes were stained with DAPI and an anti-METTL3 antibody. *n* (Control) = 28, *n* (*Mettl3ᴳᵈᶠ⁹* cKO) = 14. The white rectangle represents the area of the nucleus to be enlarged. Source data are provided as a Source data file. **i** Heatmap showing the expression levels of presentative genes in both control and METTL3-null oocytes (left) and oocytes at different growth stages (right). The representative genes were identified by the most differentially expressed genes in the SN and NSN oocytes compared to control oocytes. Source data are provided as a Source data file.

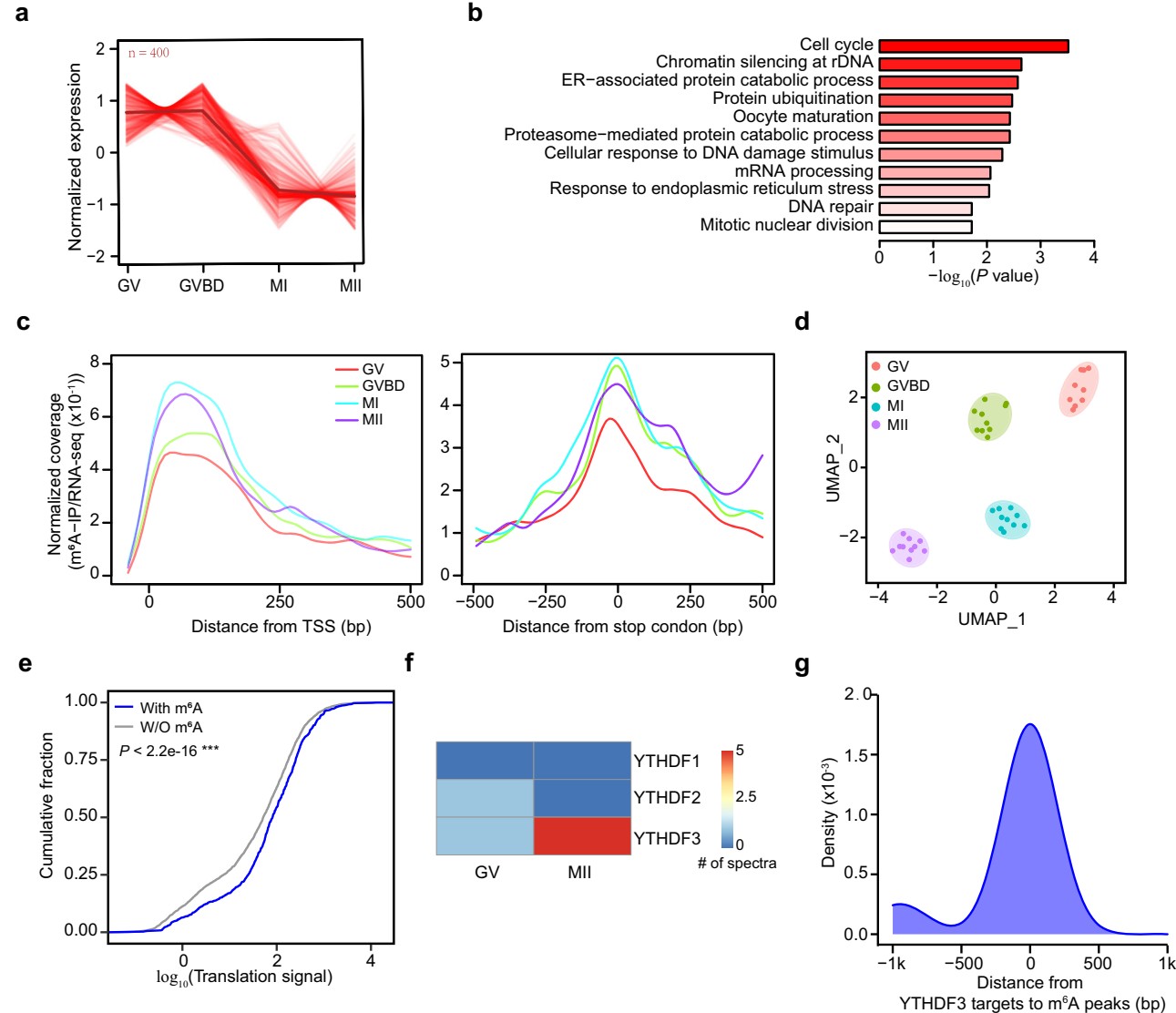

**Fig. 4 | The dynamic m⁶A landscape during oocyte maturation. a** Line plot showing the decrease in expression of the gene set during oocyte maturation as indicated by K-means clustering. **b** Bar plot displaying the enriched Gene Ontology (GO) terms in the gene set showing in (**a**). The enrichment and *P* value were calculated with default parameters using hypergeometric test of functional annotation in DAVID database. **c** Metagene profiles depicting m⁶A signals in areas surrounding the transcription start site (TSS) (left) and stop codon (right). Colored lines represent different stages of oocyte maturation. **d** Uniform manifold approximation and projection (UMAP) clustering for oocytes at different stages using the merged omics data obtained by scm⁶A-seq. **e** Cumulative frequency of translation signals[89] (GSE169632) for RNAs with or without m⁶A modification in MII oocytes. *P* values were determined by one-sided Wilcoxon test, *P* < 2.2e−16. ****P* < 0.001. **f** Heatmap displaying the protein abundance of YTHDF1/2/3 in GV and MII oocytes. **g** Density plot displaying the distance between YTHDF3 target peaks detected by enhanced crosslinking and immunoprecipitation sequencing (eCLIP-seq) in mouse embryonic stem cells (mESCs)[42] (GSE151788) and m⁶A modified RNAs in MII oocytes by scm⁶A-seq.

## m⁶A protects maternal RNAs during MII-to-zygote transition

To clarify the regulatory role of m⁶A during oocyte-to-embryo transition, we performed scm⁶A-seq on different stages of embryos: zygote (1 C), early 2-cell (E2C), mid-2-cell (M2C), late 2-cell (L2C), and 4-cell (4C). Based on the transcriptome sequencing data, we observed 3 major clusters of cells from MII to 4C stage (Supplementary Fig 5a). We further identified three RNAs clusters based on the deferentially expressed RNAs during the developmental stages, defined as maternal decay, minor ZGA, and major ZGA clusters according to their expression changes (Fig. 5a and Supplementary Data 5), which are mainly involved in translation, cell division and embryonic cleavage pathways, respectively (Supplementary Fig. 5b–d). Of note, only less than 4% of maternal decay RNAs were marked by m⁶A in MII oocytes (Supplementary Fig. 5e). Intriguingly, from MII oocytes to early embryos, m⁶A modified RNAs had a significantly increased expression (Fig. 5b),

whereas the unmodified ones showed a decreased expression (Supplementary Fig. 5f). Thus, the expression of m⁶A modified RNAs, but not the unmodified ones, tends to be increased from MII to zygote (Fig. 5c).

Intriguingly, the m⁶A-modified RNAs degrade more rapidly than the unmodified ones in early and late 2-cell stages (Supplementary Fig. 5g and Supplementary Data 6). These results support different mechanisms of m⁶A in regulating RNA metabolism in oocytes and early embryos, which has also been well discussed in the recent report[32]. To further confirm the metabolism of m⁶A-modified RNAs during MII to Zygote transition, we collected the normal control and *Mettl3*^*Gdf9* cKO MII oocytes and zygotes for total transcriptome sequencing (Supplementary Data 7). We found that in the *Mettl3*^*Gdf9* cKO samples, the RNA metabolism is largely disordered (Supplementary Fig. 5h) and the m⁶A modified RNAs in MII are rapidly degraded during *Mettl3*^*Gdf9* cKO MII-

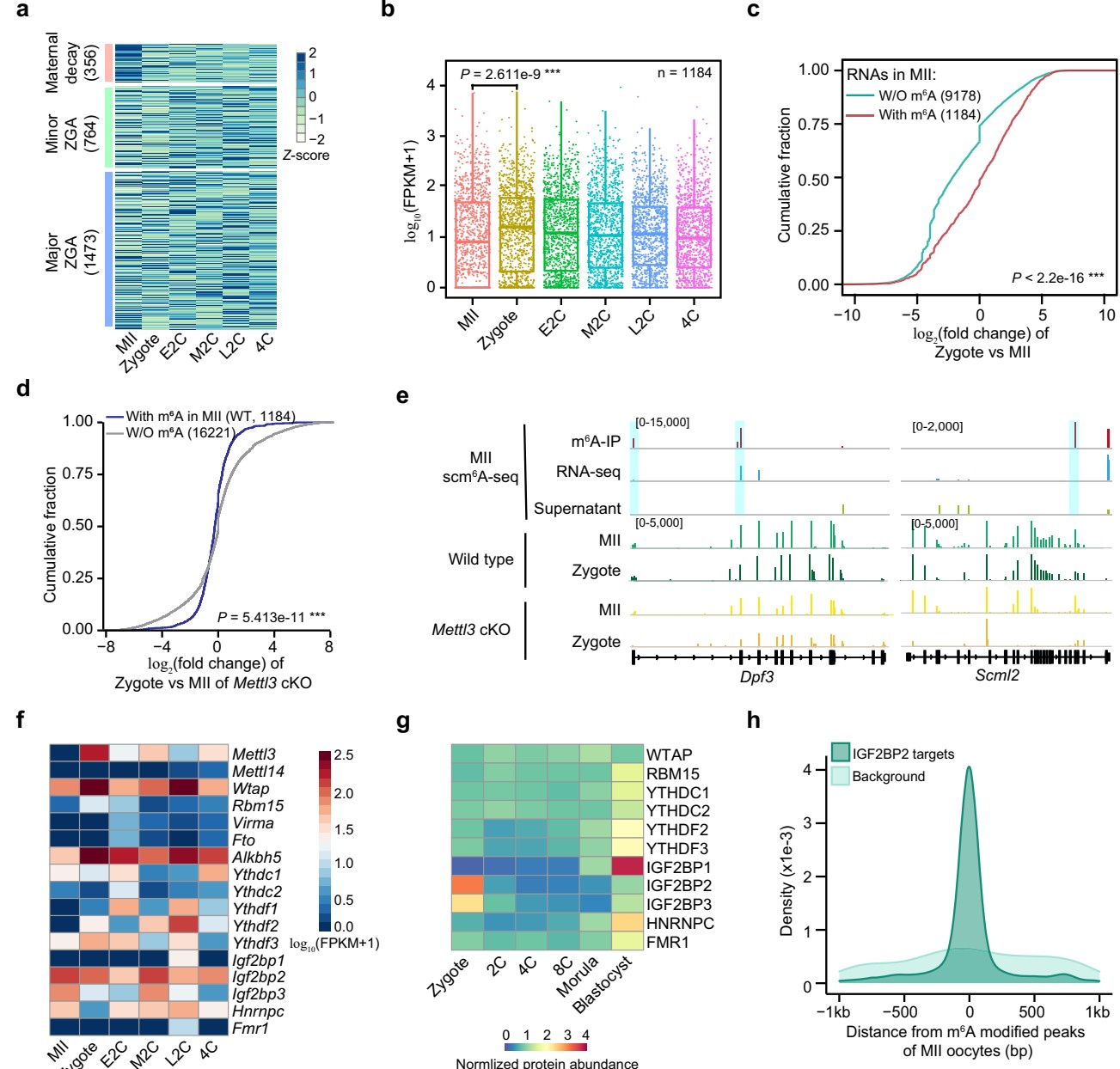

**Fig. 5 | m⁶A regulates RNA stability during the oocyte-to-embryo transition.**
**a** Heatmap showing the normalized expression of significantly dysregulated RNAs between two adjacent time points during early embryo development. The expression of maternal decay genes is the upregulated in MII oocytes, that of minor zygotic genome activation (ZGA) genes is upregulated in the early and mid-2-cell stages, and that of major ZGA genes is upregulated in the late 2-cell and 4-cell stages. **b** Box and scatter plot showing the expression of m⁶A-modified RNAs in oocytes from the MII to the 4-cell stage. The middle lines of the boxes represent the medians of datasets. The upper and bottom lines of the boxes are respectively the upper quantile and the lower quantile of the data. The whiskers mark the upper and lower limits of these datasets respectively. *P* values were determined by one-sided Wilcoxon test, *P* = 2.611e−9. ***P* < 0.001. **c** Cumulative fraction of RNA expression change (log₂(fold change)) between zygote and MII oocytes of WT oocytes. *P* values were determined by one-sided Wilcoxon test, *P* < 2.2e−16. ****P* < 0.001. **d** Cumulative fraction of RNA expression change (log₂(fold change)) between zygotes and MII oocytes of *Mettl3*^*Gdf9* cKO oocytes. *P* values were determined by one-sided Wilcoxon test, *P* = 5.413e−11. ****P* < 0.001. **e** Genome browser showing the representative RNA abundance of m⁶A modified maternal RNAs in control and *Mettl3*^*Gdf9* cKO MII oocytes and zygotes. **f** Heatmap showing the expression level of m6A-related RNAs in scm⁶A-seq data during early embryonic development. **g** Protein abundance of expressed m6A-related proteins during early embryonic development. **h** Density plot displaying the distance between IGF2BP2 target peaks detected by enhanced crosslinking and immunoprecipitation sequencing (eCLIP-seq) in human embryo steam cells (hESCs)[52] (GSE78509) and m⁶A peaks identified in MII oocytes by scm⁶A-seq.

to-zygote transition (Fig. 5d). Besides, a higher proportion of the m⁶A modified RNAs in oocytes are enriched in the hypo-stable group (Supplementary Fig. 5i), but substantially decrease during MII-to-zygote transition in the *Mettl3*^*Gdf9* cKO samples (Fig. 5e). These results suggest that m⁶A protects RNA from degradation in an METTL3-dependent manner.

To investigate the RNA binding proteins for regulation, we further evaluated the transcription and protein abundance of the m⁶A-related writers, erasers, and readers during oocyte-to-embryo transition. We observed persistent transcription of *Wtap*, *Alkbh5*, and I*gf2bp2* in the RNA-seq data of scm⁶A-seq (Fig. 5f). And IGF2BP2 protein abundance remains high at oocyte and zygote stages but decreases upon entry

into the 2-cell stage (Fig. 5g, Supplementary Fig. 5j), and whose translation signal is decreased after fertilization (Supplementary Fig. 5k). Besides, we also observed considerable co-localization between m⁶A peaks of maternal RNAs detected in MII oocytes and the binding targets of IGF2BP2 in ESCs identified by eCLIP-seq[52] (GSE78509) (Fig. 5h), suggesting that IGF2BP2, as a m⁶A reader protein, might recognize m⁶A modified maternal RNAs in the oocytes and enhance their stability during MII-to-zygote transition.

### scm⁶A-seq reveals the heterogeneity of 2-cell blastomeres
To further explore the role played by m⁶A in early embryonic development, we characterized the m⁶A signatures at single blastomere resolution and observed a trend of decreased m⁶A level after ZGA (Supplementary Fig. 6a), which is consistent with the result detected by HPLC in the previous report[32]. And the m⁶A enriched fragments showed stage specificity but no significant correlation with RNA expression (Fig. 6a, Supplementary Fig. 6b and Supplementary Data 8), pointing to stage-specific functions (Supplementary Fig. 6c). More elaborate stage-specific cell populations were observed when we introduced m⁶A signatures into a clustering assay (Fig. 6b). These results suggest a higher sensitivity and reliability of scm⁶A-seq in identification of m⁶A-level and classification of single cells.

The blastomeres of 2-cell embryos have been reported to be molecular heterogeneous and contribute unequally to the differentiation of the inner cell mass (ICM) and trophectoderm (TE) during embryonic development[25,53,54]. Moreover, asynchronous development of porcine blastomeres in the 2-cell stage has been observed[55,56]. Considering the heterogeneity of blastomeres, we further explored the m⁶A roles played in asynchronous development of mouse 2-cell embryos, and obtained two cell populations on the basis of the m⁶A and expression signatures obtained by scm⁶A-seq (Fig. 6c and Supplementary Fig. 6d) with stage information (Fig. 6d). We found that nearly 95% of early 2-cell blastomeres were in Cluster 1 (Fig. 6e), and more than 98% of the blastomeres in Cluster 2 were from mid-to-late 2-cell embryos (Fig. 6f). These data suggested that cells in Cluster 1 and 2 were in the early stage as minor ZGA and advanced stages of 2-cell as major ZGA, respectively. Furthermore, we labeled the pairing information in the 2-cell stages blastomeres (Fig. 6g) and found that a reduced proportion of paired blastomeres appeared in the same cell population during 2-cell embryos development (Fig. 6h), indicating that the heterogeneity between the two blastomeres is increased in the 2-cell embryos before major ZGA. We then analyzed the leading and lagging blastomeres in 2-cell embryos due to asynchronous ZGA (Supplementary Fig. 6e), and further defined the blastomeres in mid-to-late 2-cell embryos embedded in Cluster 1 are the lagging population and those in Cluster 2 are the leading population. Through comparing the m⁶A characteristics in these two clusters, we found a larger variance in Cluster 1 even though the difference in global m⁶A level between these two groups was insignificant (Supplementary Fig. 6f), suggesting a potential role for m⁶A in gene activation regulation.

We further compared the deferentially expressed and m⁶A-modified RNAs related to ZGA between the 2 clusters, and observed more significant changes of m⁶A signals than RNA expression (fold change >4) (Supplementary Fig. 6g). To identify the upstream regulators of these changes, we downloaded ATAC-seq data[57] to correlate chromatin accessibility with RNA expression and m⁶A signals, and identified the activated RNAs in Cluster 2 with chromatin accessibility as RNAs involved in major ZGA and RNAs in Cluster 1 that had lost accessibility as silenced maternal RNAs (Supplementary Fig. 6h, i). Further analysis showed that the most RNAs involved in the major ZGA were protein-coding RNAs, while the silenced maternal RNAs contained a larger proportion of non-coding RNAs (Supplementary Fig. 6j). Regarding that transcription factors (TFs) binding is required for the chromatin accessibility and transcription[58], we identified enriched TFs on the basis of the accessible regions of genes differentially

expressed during ZGA as described in Supplementary Fig. 6h. Importantly, we found a significantly differential m⁶A signal, but the expression level of these TF mRNAs between the two clusters did not show significant difference (Fig. 6i). These results suggest that m⁶A modification on TF mRNAs may be involved in the regulation of ZGA.

During further exploration, the involvement of most m⁶A-modified TF mRNAs during embryonic development were found to be stage-specific (Supplementary Fig. 6k). Of note, only the m⁶A-modified TF mRNAs in 4-cell stage were significantly enriched in the m⁶A modified and expressed RNA populations (FPKM > 0.5), as determined by hypergeometric test (Supplementary Fig. 6l). In addition, the m⁶A modified TF mRNAs in each stage were involved in differential pathways, for example, those in MII oocytes and zygotes were annotated into cell proliferation and gene expression processes (Supplementary Fig. 6m, n), while those at early and mid-2-cell stage were enriched in development and signaling pathways (Supplementary Fig. 6o, p). Intriguingly, m⁶A modified TF mRNAs in the 4-cell stage were significantly involved in differentiation pathways (Fig. 6j), such as *Pou5f1* (Fig. 6k), which was found to be essential for early embryogenesis[59]. Moreover, we built an interaction network of m⁶A-modified TF mRNAs with their targets from TRRUST database, and found that the TF mRNAs modified by m⁶A at 4-cell stage can target some critical factors that related to lineage differentiation, such as *Cdx2*, *Nanog*, and *Sox2* (Fig. 6l). These results indicate the involvement of m⁶A in lineage specification through TFs.

## Discussion
In this study, we developed a robust scm⁶A-seq method, which is simple, practical, and versatile tool to profile the whole transcriptomic m⁶A in single cells. In this study, the individual cells are labeled through two rounds of barcoding. Then, samples were pooled together for m⁶A enrichment using anti-m⁶A antibody IP and subject to library construction and sequenced parallelly. This strategy minimizes the batch effect and variance of enrichment efficiency among individual cells, enabling us to compare the relative m⁶A level directly among single cells. Optionally, a third-round labeling can also be performed during cDNA synthesis with barcoded TSO primers, which would allow flexible choice of labeling strategies as indicated by the sample. Recently, scDART-seq[38] based on the expression of exogenously engineered YTH-APOBEC1 was developed as the first single-cell level detection method for m⁶A profiling. The advantage of our scm⁶A-seq is to detect the m⁶A at single-cell resolution under native conditions, which provides an additional powerful tool to tackle complex biological mechanisms underlying epitranscriptomic regulation.

Currently, many single-cell multi-omics technologies were reported to detect transcriptome, genome, or epigenomic markers at the same time in single cells[60], and these tools are often used to study the regulation of gene expression by epigenomic factors. Therefore, sequencing technologies that combine the transcriptome and RNA modifications are highly desired to uncover the regulation of RNA metabolism by epitranscriptomic factors. Developing multi-omics technologies to profile transcriptome with multiple RNA methylome or translatome within one cell at the same time will be very intriguing. In addition, due to the sparsity of scm⁶A-seq data, there are some technical difficulties in the analysis of single-cell multi-omics data compared to bulk MeRIP-seq data. Therefore, data sparsity and multi-dimensional data integration analysis should be considered in the development of future algorithms and computing models.

Though there are many studies focusing on the functions of m⁶A methylome during oocyte and early embryo development, some scientific questions need to be further addressed: Firstly, only the m⁶A methylome of the fully-grown oocytes is available, and the mechanism of m⁶A in regulating maternal RNA metabolism during oocyte growth is still required to be evaluated, especially for the appearance of METTL3-independent m⁶A modified RNAs which may be catalyzed by

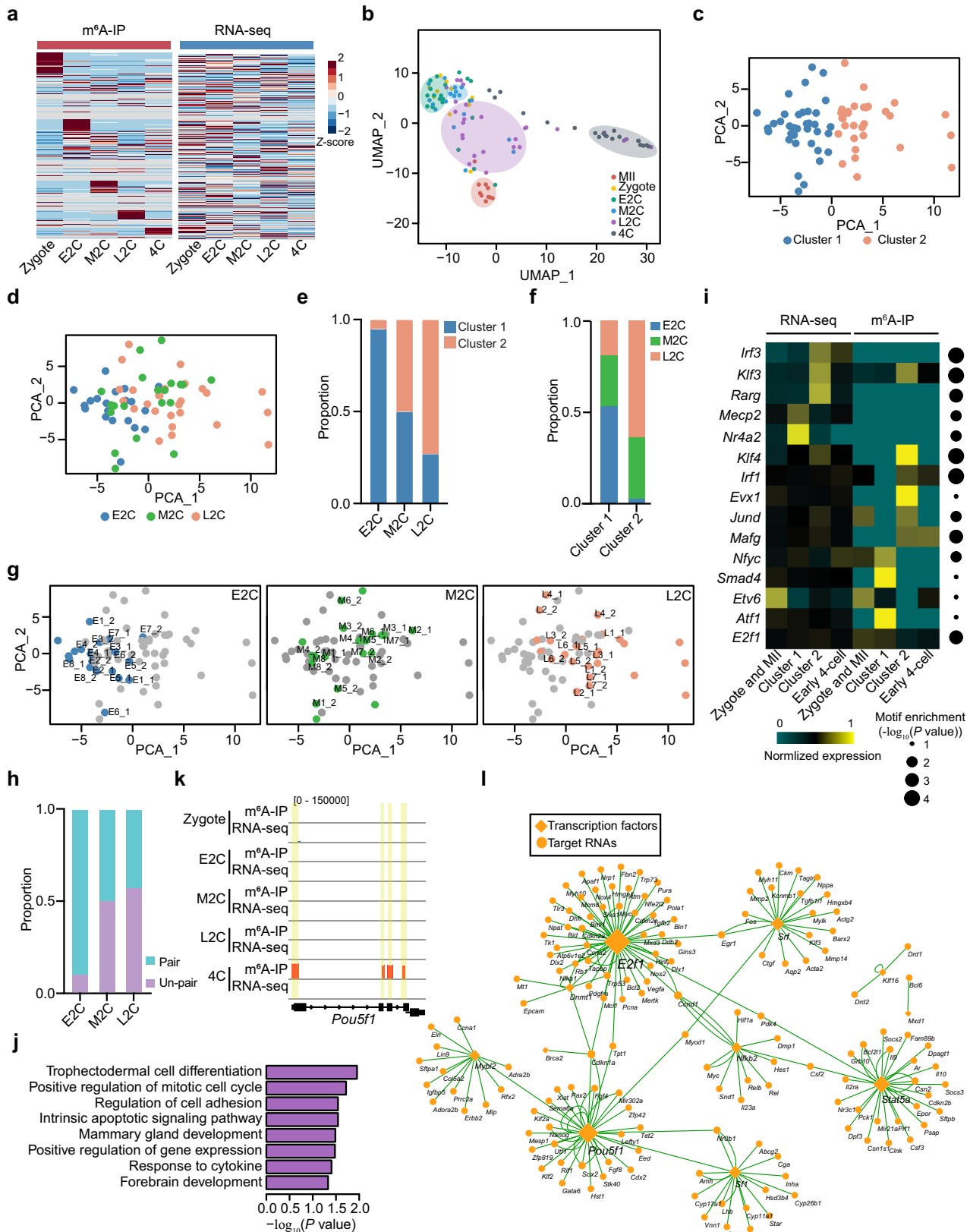

other methyltransferase such as METTL16[61]. It will be very interesting to figure out the details of newly occurred m⁶A in *Mettl3^Gdf9* cKO oocytes and the different molecular fate of m⁶A-modified RNAs during oocyte development. Secondly, our results support the conclusion that m⁶A protects the m⁶A-modified maternal RNAs from degradation during the MII-to-zygote transition (Fig. 5b–e). However, we also found many hyper-stable transcripts in the *Mettl3^Gdf9* cKO oocytes and

zygotes (Supplementary Fig. 5h), which may be caused by the decreased activity of RNA degradation machinery under the *Mettl3^Gdf9* cKO condition. Further validation is needed to fully understand the underlying mechanism, such as eCLIP-seq of different RNA binding proteins and profiling of RNA degradome. Thirdly, we found the m⁶A levels increase after meiosis resumption in the oocytes, however, the reason and biological functions of the relatively low m⁶A level in the

**Fig. 6 | Transcription factors (TFs) with m⁶A modification involved in early embryonic development. a** Heatmap displaying the expression of m⁶A-modified RNAs during early embryonic development. **b** Uniform manifold approximation and projection (UMAP) clustering of cells during early embryonic development. **c** Principal component analysis (PCA) with cells in the early, mid-, and late 2-cell stages using merged omics data of scm⁶A-seq. Cells are colored by cluster population. Source data are provided as a Source Data file. **d** Annotated stage information for the analysis presented in (**c**). Cells are colored by stage information as indicated. **e** The proportion of cells in each cluster of different 2-cell stage oocytes. **f** The proportion of cells in each stage for different clustering populations. **g** PCA plot displaying the cell pairing information of early, mid-, and late 2-cell embryos. **h** Bar plot showing the proportion of pairing information for different stages of the 2-cell embryos in two cluster populations. **i** TF mRNAs with the most enrichment motifs identified from promoter and gene body peaks of the differentially expressed genes among Cluster1 blastomeres and Cluster2 blastomeres. Source data are provided as a Source Data file. **j** Bar plot of Gene Ontology (GO) term enrichment for the TFs with m⁶A modification in the 4-cell stage embryos. The enrichment and *P* value were calculated with default parameters using hypergeometric test of functional annotation in DAVID database. **k** Genome browser showing the read abundance for *Pou5f1* in early embryo cells as indicated by the scm⁶A-seq data. The yellow rectangle boxes represent identified m⁶A peaks in 4-cell embryos. **l** TF-target network displaying the modified TF mRNAs with their targets in 4-cell stage embryos. Diamonds are the modified TF mRNAs, and the circular dots represent their targets. The size of the TF mRNAs represents the number of interacting pairs.

fully-grown oocytes remain unknown. It is now known that DNA methylation is erased from the primordial germ cells (PGCs) during their transition to primordial oocytes, and the de novo DNA methylation takes place during the follicle growth[62], but the regulation role played by m⁶A during oocyte growth and early embryo development remains unknown. It will be very interesting to explore this issue by profiling the m⁶A of PGCs and oocytes obtained from follicles at different developmental stages.

ZGA is an early hallmark of early embryonic development and may be controlled by the fertilization time, spatial location, cell cycle counts, and cell size[63]. The timing of ZGA differ among different species. In mice, minor ZGA occurs from the late 1-cell stage, and major ZGA occurs at the late 2-cell stage, and moreover, the minor ZGA is required for a normal major ZGA process[20,64]. Although the molecular bias of lncRNAs[25,53], histone modification[65], and molecules from the second polar body[66] have been studied in the 2-cell embryos, the difference of ZGA process among the individual blastomeres has not been extensively studied. Our m⁶A landscape of single blastomeres at 2-cell stage by scm⁶A-seq revealed the asynchronous ZGA process in the 2-cell embryos. This outcome can be partially explained by the different cell cycle progression between the two blastomeres in a 2-cell embryo. In addition, this is not surprising because of the appearance of many 3-cell embryos during mouse embryonic development[54,67]. The differences between the two blastomeres is a maternal trait induced by the asymmetric distribution of m⁶A methylome. As m⁶A modified RNA tend to form ribonucleoprotein (RNP) complex by m⁶A readers[68,69]. While RNPs have an important role in RNA metabolism and gene expression regulation in embryos and embryonic stem cells[70,71], we speculate that there is an asymmetric distribution of m⁶A-related RNPs in the blastomeres in the 2-cell embryos. On the other hand, the details of how m⁶A methylome regulates the ZGA process and cell fate decision need further clarification.

Overall, scm⁶A-seq provides a robust tool to explore the functions of the RNA m⁶A modification during embryo development at the single-cell level. Similar strategies might also be expanded to other RNA modifications with reliable antibodies.

## Methods
### Animals maintenance
The construction of *Mettl3* conditional knockout mice in oocytes (*Mettl3^flox/flox*;Gdf9-cre, referred to as *Mettl3* cKO) referenced from the previous report[15]. The *Mettl3^flox/flox* female mice were used as the control group (referred to as control). All mice described above were kept at C57BL/6J genetic background and housed under specific pathogen-free (SPF) conditions with 12 light/12 dark cycles at 22 °C, 40% humidity. All animal experiments were approved by the Animal Care and Use Committee of China Agricultural University.

### Collection and culture of oocytes
The control and *Mettl3^Gdf9* cKO oocytes and zygotes are collected as described before. Briefly, 4- to 6-week-old females of both control and *Mettl3^Gdf9* cKO mice were injected with 5 IU pregnant mare serum

gonadotropin (PMSG) (Solarbio, P9970). For GV oocytes, the cumulus cell-oocyte complex was collected and the oocytes (>70 μm) were released with microcapillary pipettes in M2 media 44 to 48 h after PMSG injection. For MII oocytes, the mice were then injected with 5 IU of human chorionic gonadotropin (hCG) (Ningbo No.2 hormone factory, Zhejiang, China). Then the MII oocytes were collected 14–16 h later. For zygotes, the mice were mated with male mice with known fertility, and successful mating was confirmed by the presence of vaginal plugs. For *Mettl3^Gdf9* cKO zygotes, the sample is further confirmed by the morphology of the pronuclei.

For in vitro maturation, the collected GV oocytes from 6- to 8-week-old wild-type mice were cultured in MEM medium (Gibco, 11095072) with slight modifications[72]. The GVBD oocytes and MI oocytes were collected after 4 and 8 h, respectively. The MII oocytes were collected after 12 h when the fist polar body appears.

### Embryo collection and single blastomere isolation
Embryos were collected from 4- to 6-week-old C5LNB6 females mated with C5LNB6 males. To induce ovulation, females were administered 5 IU of hCG intraperitoneally, 44–48 h post injection of 5 IU of PMSG. Embryos were collected from female mice at the following time points post hCG injection: one-cell stage (19–21 h post hCG), early two-cell stage (31–32 h post hCG), mid-two-cell stage (39–40 h post hCG), late two-cell stage (46–48 h post hCG), and four-cell stage (54–56 h post hCG). To obtain the single cell of embryos, the zona pellucida was removed using Tyrode's solution and then the embryos were treated with 0.25% Trypsin-EDTA for several seconds. The single cell was then put in the lysis buffer for scm⁶A-seq.

### Immunofluorescence staining
Oocytes were fixed in 4% PFA for 30 min at room temperature (RT) followed by permeabilization in 0.25% Triton X-100 for 15 min before blocking in 3% BSA for 1 h at RT. Then the oocytes were incubated with primary antibodies (Anti METTL3, Abcam, Cat # ab195352, 1:500 in washing solution) in 3% BSA overnight at 4 °C. The embryos were washed 3 times for 5 min each in washing solution (0.1% Tween-20 in PBS) and incubated with donkey anti-Rabbit IgG (H + L) Highly Cross-Adsorbed Secondary Antibody, Alexa Fluor 488 conjugated (Invitrogen, Cat # A21206, 1:500 in washing solution) for 1 h at RT. After washed thoroughly for three times in washing solution, embryos were stained with DAPI (1:1000 in washing solution) for 5 min and subjected to take images using a confocal microscope.

### Library preparation for scm⁶A-seq
All adenylated adapters were synthesized using a 5′ DNA Adenylation Kit (NEB, M2611A) according to the manufacturer's instructions, and stored at −20 °C for no more than six months. Single cells were piped into 96-well plates with 5 μl lysis buffer (0.5 μl of 10× lysis buffer (TaKaRa, 635013), 2.5 μl 2×FPE Buffer (Vazyme, N402), 0.5 μl of 40 U/ml RNase inhibitor, murine (NEB, M0314L), 0.5 μl of gDNA eraser from HiFiScript gDNA Removal cDNA Synthesis Kit (CWBIO, CW2582M), and

1 μl of nuclease-free water. The samples were kept at RT for 3 min for cell lysis before gDNA was removed by gDNA eraser at 42 °C for 5 min. Then, RNA was fragmented to approximately 200 nt by heating at 94 °C for 8 min. The gDNA eraser was deactivated by heating at 75 °C. Then, T4 PNK and PNK buffer (NEB, M0201 L) were added to each reaction at 37 °C for 45 min to remove the RNA 3′ phosphate group before RNA adapter ligation. For the ligation reaction, 2 μl T4 ligation buffer, 1 μl T4 RNA ligase 2, truncated K227Q (NEB, M0373L), 7 μl 50% PEG8000 (NEB, M0373L), 1 μl adenylated adapter (with barcode 1), and 1 μl RNase inhibitor (NEB, M0314L) were added to each tube. RNA ligation was performed at 4 °C overnight with gently shaking at 350 rpm. Then, RNA from all wells was pooled together and purified using 1.5× RNA Clean XP beads (Beckman, A63987) after inactivating the T4 RNA ligase by heating at 75 °C for 20 min. For the second-round labeling, the well-ligated samples were pooled together and purified with 1× RNA clean beads. The mixed samples are annealed with adapters with barcode 2 before gap filling with T4 DNA polymerase at 12 °C for 15 min according to the instructions (NEB, M0203L). The RNAs from different samples were pooled together and divided into two parts: one-fifth were retained for Input (designated as RNA-seq), the remaining four-fifths were subjected to m6A enrichment through incubating with m6A antibody and protein A-G beads according to the instructions of Magna MeRIP m6A Kit (Merck, 17-10499). After m6A immunoprecipitation (IP), the m6A-containing RNAs were eluted from the beads with $N^6$-Methyladenosine-5′-monophosphate sodium salt (designated as m6A-IP) and the supernatant were also collected and sequenced as the background for m6A detection. The RNAs from input, supernatant, and m6A-IP samples were extracted, dissolved in 40 μl nuclease-free water, and subjected to reverse transcription and the template-switching reaction using picoRT oligo and 5′ blocked TSO. The cDNAs were purified using 1× DNA clean beads and eluted in 32 μl nuclease-free water. Then, 40 μl 2× KAPA HiFi HotStart ReadyMix (KAPA, KK2602), 4 μl pico-RT primer, and 4 μl T7 primer were added for the first PCR amplification. Then, the T7-promoter-inserted libraries were transcribed into RNAs with T7 polymerase and further purified using RNA clean beads (Vazyme, N412-01). Then, rRNA products were removed using an RNase H-based method as described previously[73]. Then, the rRNA-minus RNAs were annealed and reverse transcribed using pico-RT primer. Finally, the cDNA was amplified using sequencing adapters (NEB, E7335L) before library purification with 1× DNA clean beads (Vazyme, N411-01). Library sequencing was performed on the Illumina NovaSeq6000 system. Notably, for the 10 individual GV oocytes as data quality control, the m6A-containing RNAs were extracted by proteinase K digestion instead of competitive elution after m6A-IP. All the oligos used were ordered from GeneScript. The oligo sequences were provided in Supplementary Data 9.

## Library construction for bulk MeRIP-seq and RNA-seq

For bulk MeRIP-seq and RNA-seq of 25, 50, and 70 GVs, the RNA was extracted using RNeasy Plus Kits (Qiagen, 74034) according to the instruction ignoring the DNA filter step. Then the total RNA was subjected to m6A enrichment according to procedures of scm6A-seq except for without cell labeling. After m6A immunoprecipitation, the m6A-Protein A/G beads-RNA complexes were washed first by low salt buffer followed by high salt buffer and m6A binding buffer. Then the m6A-modified RNA fragments were extracted by proteinase K digestion and acid phenol-chloroform as described previously[15]. The library construction was also performed as for scm6A-seq. Sequencing was performed on an Illumina NovaSeq 6000 platform.

## scm6A-seq data preprocessing

The template switch oligo (TSO) sequence of raw sequencing reads was trimmed by Cutadapt[74] (version 2.10) with parameters '-g TCCGATCT -O8 -n5 -e 0 --action trim', '-g CACGTCTC' for multiple TSO sequences filtering, and multiple adapter sequences trimming with

parameters '-a NNNNNNNAGATCGGA -a TCGGAAGAGCACAC -A NNN NNNNAGATCGGA -A TCGGAAGAGCACAC -e0 -j 6 -m 28 -O8'. Then reads of individual cells were separated by barcode sequences (Supplementary Data 10) using a homemade script. First, the reads starting with "[barcode]NNNNGG" were selected, and other reads were eliminated. For single sample data, the four-base random sequence and the first seven bases at read 2 were combined and wrote at the first line of the fastq-format and defined as unique molecular identifiers (UMIs). To remove the PCR duplicate reads, the first seven bases at the beginning of read 1 and read 2 were combined together as a molecular barcode. Then, the molecular barcodes were sorted, and the unique reads were retained. For single-cell parallel sequencing data, the index barcode, barcodes at read 1 and read 2 were decoded step by step. The data are separated into individual files, corresponding to each single cell. All scripts used in this study are available from the corresponding author by request.

After single-cell data separation, the reads of m6A-IP, supernatant, and RNA-seq from scm6A-seq and bulk sequencing were aligned to the *Mus musculus* (mouse) genome (GRCm38/mm10) from Ensembl[75] (release 96) using HISAT2[76] (version 2.0.5). Samtools (version 1.6)[77] was used to filter and sort the mapped reads, then featureCounts (version 2.0.3)[78] program was applied to summarize the gene counts in each sample, and fragments per kilobase per million mapped reads (FPKM), transcripts per kilobase of exon model per million mapped reads (TPM) and counts of exon model per million mapped reads (CPM) genes expression were calculated by StringTie (version 2.2.1)[79]. The relative m6A level of each cell is defined as $\log_2$(reads count (m6A-IP/(m6A-IP + RNA-seq + Supernatant))).

## Saturation analysis for scm6A-seq data

Sequencing saturation of scm6A-seq was evaluated by random re-sampling approach. The aligned bam files of all reads in each GV oocyte were annotated to mm10 genome by BEDTools (version 2.30.0)[80]. Then, shuf of Linux was used to randomly select a given number of reads from total reads and the number of mapped reads covered genes was counted. The given number of reads and corresponding covered genes were applied to build saturation curve for scm6A-seq.

## Metagene analysis for scm6A-seq data

The relative position of each transcription start site (TSS) and stop codon in the corresponding transcript were identified in Ensembl mm10 genome (release 96), respectively. For metagene density plots, bigwig files were converted from bam files using bamCoverage of deepTools[81] (version 3.3.1) with 5 bp bins, then computeMatrix command was used to calculate scores per genome region with the parameter 'reference-point --referencePoint TSS', and '-b 500 -a 500 -R ${StopCondon}' for the stop codon or '-b 50 -a 500 -R ${TSS}' for TSS sites in each transcript. Finally, the ratio of m6A-IP to RNA-seq from scm6A-seq in the score matrix files was calculated and visualized by density plot. To generate the metagene profiles for each stage, Samtools was used to merge single-cell data from scm6A-seq.

## Identification of high-confident m6A peaks

MACS2 callpeak[82] (version 2.1.4) was used for m6A-enriched peaks calling for m6A-IP with corresponding RNA-seq as control in scm6A-seq and bulk sequencing. The parameters of '--keep-dup all --nomodel -g mm -B -q 0.05' were used for peak calling. Furthermore, high-confident m6A peaks were identified when peaks were annotated to transcriptome of Ensembl mm10 (release 96) with 'fold_enrichment > 2 && $-\log_{10}$(q value) > 10' in the peak calling files.

For scm6A-seq, only the peaks identified in both RNA-seq and supernatant as control samples were considered high-confident and used for further analysis by intersectBed of BEDTools (version 2.26.0) with '-f 0.51 -F 0.51'. Merged data was used as input samples for

identification of m⁶A peaks in each stage. METTL3-dependent m⁶A peaks in GV oocytes were defined as enriched peaks only identified in control compared with the *Mettl3^Gdf9* cKO samples with expression more than 1 (FPKM ≥ 1).

For bulk[15] and low input[32] m⁶A sequencing data, the parameters of '--keep-dup all --nomodel -g mm -B -q 0.05' were used for peak calling. The m⁶A peaks were then annotated to transcriptome of Ensembl mm10 (release 96) for the identification of the m⁶A-modified RNAs in each replicate.

### m⁶A motif enrichment analysis

Motif discovery for filtered m⁶A peaks was identified by findMotifs-Genome.pl of HOMER software (version 4.11.1). The sequences extracted inside the m⁶A peaks from the transcriptome was used as target sequences and the randomly shuffling peaks upon whole transcriptome were obtained by shuffleBed of BEDTools.

### Differential expression of METTL3-dependent m⁶A modified RNAs

Differentially expressed genes between *Mettl3^Gdf9* cKO and control oocytes were calculated by edgeR[83] with the $P$ value <0.05. For further confirm, Integrative Genomics Viewer (IGV)[84] was used to display the m⁶A signals and RNA abundance of scm⁶A-seq in mm10 genome with bigwig files generated by BEDTools.

### Clustering analysis of scm⁶A-seq data

Before clustering analysis, gene expression from single-cell and merged samples of scm⁶A-seq among different oocytes and embryo developmental stages was $z$ score-normalized. Clustering of gene sets during oocyte maturation and early embryonic development was performed by K-means algorithm[85] with 10,000 iterations. Hierarchical clustering analysis was applied to classify cell samples using hclust function in R program. The principal component analysis (PCA) and uwot of uniform manifold approximation and projection (UMAP) method[86] was used for dimensionality reduction and cell clusters of single-cell data. For further clustering of the scm⁶A-seq data by comprehensive analysis of the gene expression level and RNA m⁶A level, the single cell matrix after dimensionality reduction by UMAP using monocle2[87] and the $\log_2((\text{IP reads})/(\text{Total Reads}))$ values were used for hierarchical clustering.

### m⁶A reader analysis

YTHDF2/3 binding peaks identified by eCLIP-seq in mESCs were downloaded from GEO database (GSE151788)[42]. IGF2BP2/3 binding peaks of hESCs were extracted from eCLIP-seq data (GSE78509)[52]. UCSC liftOver[88] was used to convert genome coordinates between human and mouse. shuffleBed of BEDTools was applied to randomly permute the genomic locations of m⁶A reader binding peaks. Distance between m⁶A readers' targets and identified m⁶A peaks was calculated using the "closet" command of BEDTools.

### Translatome and proteome analysis

The translation signals and efficiency of different developmental stages for each gene were calculated from public data (GSE169632)[89]. Translation signals were defined as gene expression levels calculated from LiRibo-seq data. Translation efficiency (TE) of m⁶A-modified RNAs was evaluated by ratio of RPKM between LiRibo-seq and RNA-seq samples in different developmental stages. The protein abundance of oocyte maturation, mESCs, and early embryonic developmental stages were obtained from LC-MS/MS[51] and TMT-based quantitative MS assay data[90], respectively.

### Maternal and ZGA genes analysis

Genes expressed in MII with FPKM ≥ 1 were considered as maternal genes. Maternal decay genes were identified by comparing RNA-seq in

scm⁶A-seq of MII and zygote using threshold of $\log_2(\text{fold change}) > 0.5$ and $P$.adj < 0.05 in DESeq2[91] with count tables from featureCounts. Minor ZGA genes were identified as genes upregulated in early and mid-2-cell stages with the previous adjacent developmental time point as control ($\log_2(\text{fold change}) > 0.5$ and $P$.adj < 0.05). Major ZGA genes were defined by comparing the late 2-cell and 4-cell embryos with $\log_2(\text{fold change}) > 0.5$ and $P$.adj < 0.05 to samples of previous adjacent developmental time point, respectively. K-means algorithm was also used for clustering of maternal and ZGA genes during early embryonic development.

### Functional enrichment analysis

For the functional annotation of gene sets, DAVID[92], Matascape (version 3.5)[93], and ClueGO[94] of cytoscape[95] were applied to annotate Gene Ontology Consortium[96] biological processes with threshold $P$ value <0.05 without special instructions.

### Transcriptome factors analysis

Transcription factors and their corresponding targets in mice were collected from TRRUST database[97] (version 2), and their interaction network was visualized by Cytoscape (version 3.9.1). UpsetR (version 1.4.0)[98] was applied to display the overlapped RNAs among different developmental stages.

To analyze the TFs responsible for ZGA. The differently expressed genes are further filtered with threshold $\log_2(\text{FPKM(Class2)}/\text{FPKM(Class1)}) > 2$. Then the sequence inside ATAC-seq peaks in the promoter region and gene body of these genes are used for TFs enrichment using MEME-Chip (version 5.5). The ATAC-seq peaks are identified using public data (GSE66581) of the 2-cell embryos using MACS2 with default parameters.

### Statistics and reproducibility

All statistical analyses were performed using R program and Prism software. Statistically significant differences between different groups were evaluated by one- or two-sided test (Wilcoxon test, Hypergeometric test, or Student $t$ test) without additional adjustments. All significant levels are presented in figures as singe asterisk (*) when $P$ value < 0.05, **$P$ < 0.01, and ***$P$ < 0.001, unless otherwise specified. More than 10 cells of each stage of oocytes/embryos are subjected to scm⁶A-seq. And more than 4 oocytes of each stage were collected for single-cell RNA-seq.

### Reporting summary

Further information on research design is available in the Nature Portfolio Reporting Summary linked to this article.

## Data availability

The raw sequencing data generated in this study have been deposited in Genome Sequence Archive of National Genomics Data Center under accession code CRA006425. The expression value, m6A peaks, and other necessary matrix data generated in this study are provided in the Supplementary Information and Source Data files. The immunofluorescence staining (IF) data generated in this study have been deposited in Mendeley (https://data.mendeley.com/datasets/dhskwr3r5f/draft?a=69360c62-93e5-415a-b5cf-36b51dcfe0f9). Source data are provided with this paper.

## Code availability

The main code and demo test data have been deposited in Zenodo (https://zenodo.org/record/6809317#.YsfA0IRByUk). Other codes are available from the lead contact upon request.

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

## Acknowledgements

This work was supported by grants from the Innovative Research Group Project of the National Natural Science Foundation of China (32121001 Y.G.Y.), the Strategic Priority Research Program of the Chinese Academy of Sciences, China (XDA16010501 Y.G.Y., XDB38020500 Y. Y., XDPB2004 Y.Y.), the CAS Project for Young Scientists in Basic Research (YSBR-073 Y.G.Y.), National Key R&D Program of China (2019YFA0110900 Y.P.S., 2019YFA0110901 Y.G.Y., 2019YFA0802201 Y.S.C.), and the National Natural Science Foundation of China (92053115 Y.Y., 91940304 Y.Y.), CAS Key Research Projects of the Frontier Science (QYZDY-SSW-SMC027 Y.G.Y.), the Youth Innovation Promotion Association of CAS (2018133 Y.Y.), the Beijing Nova Program (Z201100006820104 Y.Y., 20220484210 Y.Y.), and the Shanghai Municipal Science and Technology Major Project (2017SHZDZX01 Y.G.Y.), and the K. C. Wong Education Foundation (Y.G.Y.).

## Author contributions

H.Y. performed the experiments with the help of J.W.X., G.G.S., Y.J.G., and D.B.L.; D.R.Z. and Q.L. collected the oocytes and embryos. Y.T.C. helped to check the chromatin configuration of GV oocytes; C.C.G. performed bioinformatics analysis with critical input from X.F. and Y.S.C.; Y.G.Y., Y.P.S., Y.Y., and J.Y.H. conceived this project, supervised the study and interpreted the data. Y.G.Y., Y.Y., C.C.G., Y.L.Z., L.H., H.Y., wrote the manuscript, and with contributions from all the authors.

## Competing interests

The authors declare no competing interests.
