## [Peer Review File · Nature Communications]

scm6A-seq reveals single-cell landscapes of the dynamic m6A during oocyte maturation and early embryonic developmentREVIEWER COMMENTS

Reviewer #1 (Remarks to the Author):

In this manuscript entitled "scm6A-seq reveals single-cell landscapes of the dynamic m6A during oocyte maturation and early embryonic development", Yao et al., revealed dynamic profiles of m6A in single cell resolution during oocytes maturation and early embryonic development using scm6A-seq, which can capture the single cell information in NSN and SN GV oocytes and blastomeres of 2-cell. Authors revealed potential role of m6A on degradation of development-silencing RNAs during oocytes development and maturation using Mettl3 cKO oocytes. Moreover, authors found the m6A-dependent asymmetries in blastomeres of 2-cell embryos. Overall, the study is challenging and interesting with several important new findings.

Major comments:

1. In Fig2, author defined 1214 Mettl3-dependent modified RNAs in GV oocytes. How many Mettl3-independent modified RNAs in GV? And what's the expression level change of Mettl3-independent modified RNA after Mettl3 depletion? It's better to include the Mettl3-independent modified RNA as control group in fig2d.
2. The conclusion in Fig5 that m6A protects its modified RNAs from degradation during MII to zygote transition is not such strong in Fig5b and 5c. Can you use the RNA-seq data of Mettl3-cKO oocytes which will arrest in zygote to supplement the data?
3. The content of the whole manuscript are too much and complicated. Some information about the expression profile such like line 249-264, 285-296 may be removed to highlight the major points. The subhead of each paragraph should be revised to strengthen the conclusion.

Minor comments:

1. In Fig1f, the label of y-axis showed (%), which seems to be removed according to the text.
2. Line 145-146, add a brief description of the phenotype of Mettl3 cKO oocytes.
3. In sup Fig3g, the control group are the total RNAs, it's better to choose random RNAs with the same gene number as development-silencing RNAs.
4. The order of Fig4e and Fig4g seems to be wrong according to the text.
5. In Fig5b, how many genes are plotted? Number should be showed also in Sup Fig5g.
6. In Sup Fig6a, the variance among cells at the same stage seems too large to get the conclusion that the m6A level are decreasing after zygote.
7. The recent publication in Nature Cell Biology should be cited.

Reviewer #2 (Remarks to the Author):

Advance summary and potential significance to field

In this study, Yao and colleagues develop a new technique of single-cell m6A sequencing (scm6A-seq) to profile the m6A methylome and transcriptome simultaneously. The authors observe the dynamic changes of m6A modified RNAs during oocyte maturation and early embryonic development. They find that m6A deficiency led to aberrant RNA clearance and low quality of oocytes in the model of Mettl3 conditional knockout mouse. The asymmetric epi-transcriptome that is dependent on m6A further explains the heterogeneity between the blastomeres of 2-cell embryo. Overall, this study is carried out by the method of scm6A-seq in the single cell dimension to investigate in-depth into m6A characteristics and functions at the stage of gametogenesis and early embryonic development.

Comments for the author

However, before publication, further interpretation and discussion of the data as well as additional

experiments would be necessary to support the authors' conclusions.

Major comments:

1. Fig. 1D shows that the m6A modified fragments detected by scm6A-seq are the highest enrichment ratio in CDS region, which was not consistent with the previous reports that m6A was mainly found in 3'UTR, as mentioned in the Introduction (line 53). Please explain this difference.
2. As shown in Supplementary Fig. 1e, more than 10000 m6A modified RNAs are detected in the groups of 50 and 70 GVs by bulk MeRIP-seq, while, the mounts of modified RNAs are about 5000 in the group of 25 GVs by the same way and so it is with single-cell through scm6A-seq. How to achieve the optimal balance between cell number and detection amount?
3. Some statements lack for sufficient evidence in the article, for example, Supplementary Fig. 1h suggests that scm6A-seq is more sensitive than MeRIP-seq through the expression and modification of Hist4h4 gene; the integrated genomics viewer tracks of Lars2 and Mtus1 show the commonly and specifically m6A modified RNAs at different oocyte stages in Supplementary Fig. 4e, but it does not reflect the "far more" described in line 288. Please provide more evidence to support these descriptions.
4. At present, some studies have detected m6A modified RNAs at the developmental stages of oocyte and pre-implantation embryos, such as references 6 and 10 cited in this paper. Has the scm6A-seq data gained from this study been compared with these publicly available data?
5. As observed in the study, m6A modification plays the opposite regulatory role at different stages, promoting mRNA degradation in GV oocytes while protecting the modified RNAs from degradation during MII-to-zygote transition. This phenomenon is interesting, please explain it further.
6. I would like the authors to reconsider the Discussion. For example, some existing studies on RNA m6A modification in the field of reproduction should be included in the Introduction section in Introduction. In addition, most of Discussion are merely repetition or summary of the results.

Minor comments:

1. The ordinate of Fig.1F is marked incorrectly.
2. Line 157-158 has a syntax problem.
3. Fig. 2e does not reflect the role of mettl3, although it is described in the legend, please describe it as consistently as Fig. 4e
4. Fig. 3g shows that the RNA abundance in NSN group is lower than that in the SN group. Which kind of GV oocyte was selected for scm6A-seq detection in the previous section?
5. Is the order of Fig. 4e and Fig. 4G reversed?

Revision Summary

1. Summary of major comments from Editors and Reviewers.

Based on the requests and comments from the reviewers, we have performed further analyses and designed additional experiments that have largely improved our manuscript. The revised parts have been marked with red color in our revised manuscript and figures. The detailed revision results are summarized and listed in the following **Table 1**.

Table 1. Revision results for major comments from the reviewers.

Key Questions	Reviewers	New experiments/analyses
1 The expression level change of Mettl3-independent modified RNA after Mettl3 depletion.	1#: Q1	To evaluate the expression change of METTL3-independent m ⁶ A modified RNAs after Mettl3 depletion, we have firstly filtered 857 expressed (FPKM > 1) METTL3-independent m ⁶ A modified RNAs, and observed a globally increased expression of METTL3-independent m ⁶ A modified RNAs than the unmodified RNAs in Mettl3^{Gdf9} cKO oocytes (Rebuttal Fig. 1a; Revised Fig. 2d). The globally increased expression of the METTL3-independent m ⁶ A modified RNAs may be due to the association of m ⁶ A with RNA abundance. Thus, we further evaluated the abundance of both m ⁶ A modified and unmodified RNAs in wild-type and Mettl3^{Gdf9} cKO oocytes, and found that the m ⁶ A modified RNAs displayed higher expression level than the unmodified ones (Rebuttal Fig. 1b, Revised Supplementary Fig. 2d), which is consistent with the previous study performed in single cells ¹ . In addition, we found that the regulation of m ⁶ A in promoting RNA degradation in GV oocytes was also observed in Kiaa1429 , one known component of the m ⁶ A methyltransferase complex, cKO oocytes ² (Rebuttal Fig. 1c). Collectively, these data demonstrated that m ⁶ A modification contributes to RNA degradation in mouse oocytes.
2 Explain further the phenomenon that m⁶A promotes RNA degradation in the GV oocytes, while protects RNA from degradation during MII-to-	1#: Q2 2#: Q5	We have observed that Ythdf2 , which mediates the degradation of m ⁶ A modified RNAs, expressed relatively higher level in the growing oocytes than the fully-grown oocytes (Revised Supplementary Fig. 3e) based on the public data ³ , which supports by the previous publication ⁴ about the expression of YTHDF2 in the growing oocytes.

	zygote transition		Besides, we have observed a relative higher translation signal of Ythdf2 in GV than MII oocytes and zygotes (Rebuttal Fig. 8a, Revised Fig. 2e) using public data⁵. Combined with the public eCLIP-seq data⁶, we also found the METTL3-dependent m⁶A modified peaks are close to the YTHDF2 targets (Revised Fig. 2e) and more stable in Mettl3^{Gdf9} cKO oocytes (Revised Fig. 2c). These results suggest that YTHDF2 is the probable regulator for the m⁶A mediated RNA degradation in the GV oocytes. For the relative higher stability of m⁶A modified RNAs during MII-to-zygote transition, we have performed translation analysis and found an obvious decrease in the translation signal of Ythdf2 (Rebuttal Fig. 8a, Revised Fig. 2e), but a continued robust translation of Igf2bp2 from GV to MII stage⁵ (Rebuttal Fig. 8b, Revised Supplementary Fig. 5k), suggesting that IGF2BP2 may play a major role in stabilizing the methylated RNAs in MII oocytes. Furthermore, we have collected both control and Mettl3^{Gdf9} cKO MII oocytes and zygote embryos and performed total transcriptome sequencing. Interestingly, we have found that the RNA metabolism is largely dysregulated from the oocytes of Mettl3^{Gdf9} cKO to zygotes (Rebuttal Fig. 8c, Revised Supplementary Fig. 5h), and a greater proportion of m⁶A modified RNAs fall in the category of hypo-stable RNAs (Rebuttal Fig. 8d, Revised Supplementary Fig. 5i), suggesting that m⁶A contributes to RNA degradation in the Mettl3^{Gdf9} cKO oocytes (Rebuttal Fig. 8e, Revised Fig. 5e). These results were consistent with the finding that m⁶A increases the stability of m⁶A modified mRNAs in normal MII oocytes during MII-to-zygote transition.
3	Explain the m ⁶ A enrichment difference in CDS or 3'UTR.	2#: Q1	We have summarized the m ⁶ A distribution in different relevant publications and reanalyzed the m ⁶ A distribution with stop codon regions included and the result showed the m ⁶ A distribution in our work is consistent with previous publications (Rebuttal Fig. 4, Revised Fig. 1d).
4	How to achieve the optimal balance between cell number and detection amount?	2#: Q2	We have evaluated the correlation of m ⁶ A detection amount and cell number or genome coverage, and observed a positive correlation between the number of detected m ⁶ A peaks with the number of cells and sequencing depth in single cell and bulk sequencing ⁷ (Rebuttal Fig. 5). Therefore, there is not a simple conclusion about the balance between the number of cells and the detection amount of m ⁶ A, which has a lot to do with the number of

			cells, genome coverage and analysis pipelines, etc. We have included the discussion in the Revised manuscript (Page 13, lines 383-387) .
5	The statements about the sensitivity of scm⁶A-seq and the stage specific m⁶A methylome lack sufficient evidence in the article.	2#: Q3	To further illustrate the sensitivity of scm⁶A-seq for m⁶A detection, we have compared scm⁶A-seq and bulk MeRIP-seq for GV oocytes and found that nearly 70% methylated RNAs in 2500 GVs bulk MeRIP-seq were also identified in scm⁶A-seq (Rebuttal Fig. 6a). Besides, there were 50% methylated RNAs only detected in single GV by scm⁶A-seq. This result suggests that scm⁶A-seq is a sensitive approach to detect both the highly conserved and low abundant m⁶A modification which can most reflect the cellular heterogeneity (Rebuttal Fig. 6b,c and Revised Fig. 1g,h). In addition, we have modified the description about stage-specific modification during oocyte maturation from GV to MII oocytes in the Revised manuscript (Pages 8-9, lines 244-249), and reanalyzed and included the information about several more stage-specific modified RNAs (Rebuttal Fig. 6d-g, Revised Supplementary Fig. 4g).
6	The comparison of the scm⁶A-seq data gained from this study with the publicly available data.	2#: Q4	To compare the scm⁶A-seq data with previous relevant publications, we have analyzed bulk MeRIP-seq data for GV oocytes with two replicates⁸, and low input ULI-MeRIP-seq data for GV oocytes with three replicates in another recent publication⁷. We found that the modified RNAs detected by scm⁶A-seq covered nearly 70% of the modified RNAs from bulk MeRIP-seq and 55% of the modified RNAs from ULI-MeRIP-seq (Rebuttal Fig. 7a, Revised Supplementary Fig. 1g), and moreover, the 1746 conserved methylated RNAs were enriched in the critical biological processes for GV oocytes (Rebuttal Fig. 7b, Revised Supplementary Fig. 1h). Furthermore, we found that many vital mRNAs, such as E2f1 and Bod1, are modified by m⁶A and are highly conserved among single cells (Rebuttal Fig. 7c, Revised Supplementary Fig. 1i).
7	Strengthen the subheading and summary conclusion.	1#: Q3 2#: Q6	We have reorganized the subheadings to strengthen the conclusion for each part of results in the Revised manuscript (Page 6, line 150; Page 7, line 188; Page 8, line 230; Page 9, line 266; Page 10, line 303). Besides, we also revised the Discussion section in the Revised manuscript (Page 13, lines 383-401).

2. Point-by-point responses to reviewers

Reviewer #1 (Remarks to the Author):

In this manuscript entitled “scm6A-seq reveals single-cell landscapes of the dynamic m6A during oocyte maturation and early embryonic development”, Yao et al., revealed dynamic profiles of m6A in single cell resolution during oocytes maturation and early embryonic development using scm6A-seq, which can capture the single cell information in NSN and SN GV oocytes and blastomeres of 2-cell. Authors revealed potential role of m6A on degradation of development-silencing RNAs during oocytes development and maturation using Mettl3 cKO oocytes. Moreover, authors found the m6A-dependent asymmetries in blastomeres of 2-cell embryos. Overall, the study is challenging and interesting with several important new findings.

Major comments:

1. In Fig2, author defined 1214 Mettl3-dependent modified RNAs in GV oocytes. How many Mettl3-independent modified RNAs in GV? And what's the expression level change of Mettl3-independent modified RNA after Mettl3 depletion? It's better to include the Mettl3-independent modified RNA as control group in fig2d.

Response: Thanks for these thoughtful comments. To evaluate the expression change of METTL3-independent m⁶A modified RNAs after *Mettl3* depletion, we have firstly filtered 857 expressed (FPKM > 1) METTL3-independent m⁶A modified RNAs, which were defined as only appeared in *Mettl3*^{Gdf9} cKO GV oocytes, and then compared their expression fold-change with METTL3-dependent and unmodified (W/O m⁶A) RNAs. Except for the METTL3-dependent m⁶A modified RNAs, we also observed a globally increased expression of METTL3-independent m⁶A modified RNAs than the unmodified RNAs in *Mettl3*^{Gdf9} cKO oocytes (**Rebuttal Fig. 1a, Revised Fig. 2d**). For the METTL3-dependent m⁶A modified RNAs, we have elucidated that these RNAs tend to be degraded in GV oocytes (**Revised Fig. 2g; Revised Supplementary Fig. 2e,f**), thereby showing globally increased expression in *Mettl3*^{Gdf9} cKO oocytes. For the METTL3-independent m⁶A modified RNAs, the globally increased expression may be due to the association of m⁶A with RNA abundance. Thus, we have further evaluated the abundance of both m⁶A modified and unmodified RNAs in wild-type and *Mettl3*^{Gdf9} cKO oocytes, and found that the m⁶A modified RNAs display a higher expression level than the unmodified ones (**Rebuttal Fig. 1b, Revised Supplementary Fig. 2d**), which is consistent with the previous study performed in single cells using scDART-seq¹. In addition, we have also analyzed the data from a previous study on KIAA1429², one known component of the m⁶A methyltransferase complex, and found that the regulation of m⁶A in promoting RNA degradation in GV oocytes was also observed in *Kiaa1429* cKO oocytes (**Rebuttal Fig. 1c**).

Collectively, these data demonstrated that m⁶A modification contributes to RNA degradation in mouse oocytes. According to the current reports, the m⁶A modification could be catalyzed by different methyltransferases, including METTL3 and METTL16⁹.

Rebuttal Fig. 1. m⁶A promotes RNA degradation during oocyte development.

a. Cumulative fraction of expression changes upon *Mettl3*^{Gdf9} cKO for METTL3-dependent m⁶A modified RNAs (red), METTL3-independent m⁶A modified RNAs (green), and the unmodified RNAs (grey). *P* values were determined by Wilcoxon test. **b.** Boxplot showing the expression of METTL3-dependent and METTL3-independent m⁶A modified RNAs in wild type (green) and *Mettl3*^{Gdf9} cKO oocytes (red). **c.** Cumulative fraction of expression changes upon *Kiaa1429* cKO for KIAA1429-dependent m⁶A modified RNAs (red), KIAA1429-independent m⁶A modified RNAs (blue), and the unmodified RNAs (grey). *P* values were determined by Wilcoxon test.

2. The conclusion in Fig5 that m⁶A protects its modified RNAs from degradation during MII to zygote transition is not such strong in Fig5b and 5c. Can you use the RNA-seq data of *Mettl3* cKO oocytes which will arrest in zygote to supplement the data?

Response: Thanks for this thoughtful suggestion. To comprehensively analyze the regulation of m⁶A on RNA metabolism during MII to zygote, we have first analyzed the expression changes of the unmodified RNAs and found that compared to the increased expression level of m⁶A modified RNAs (Original Fig. 5b, Revised Fig. 5b), the unmodified RNAs showed a significantly decreased expression from MII to zygote (Rebuttal Fig. 2a, Revised Supplementary Fig. 5f). Furthermore, we have followed the reviewer’s suggestion and collected the *Mettl3*^{Gdf9} cKO MII oocytes and zygotes for total RNA-seq. Strikingly, we found that the m⁶A modified RNAs (Original Fig. 5b, Revised Fig. 5b) degrade more rapidly compared to the unmodified ones in *Mettl3* deficient MII oocytes and zygotes (Rebuttal Fig. 2b, Revised Fig. 5d). We have also included this information in the Revised manuscript (Page 10, lines 279-291).

Rebuttal Fig. 2. m⁶A protects m⁶A modified RNAs from degradation during the MII-to-zygote transition.

a. Box and whisker plot showing the expression of unmodified RNAs in oocytes from MII oocyte to 4-cell stage. *P* values were determined by Wilcoxon test. b. Cumulative fraction of expression changes of m⁶A modified RNAs (blue) and unmodified (grey) RNAs in *Mettl3*^{Gdf9} cKO MII oocytes and zygotes. *P* value was determined by Wilcoxon test. *, *P* < 0.05; **, *P* < 0.01; ***, *P* < 0.001.

3. The content of the whole manuscript are too much and complicated. Some information about the expression profile such like line 249-264, 285-296 may be removed to highlight the major points. The subhead of each paragraph should be revised to strengthen the conclusion.

Response: Thanks for these suggestions. We have followed the reviewer's advice to remove the content of expression profile of early embryo development and to highlight the major points in the **Revised manuscript (Pages 9-10, lines 267-291)**. Additionally, we have reorganized the subheadings to strengthen the conclusion for each part in the **Revised manuscript (Page 6, line 150; Page 7, line 188; Page 8, line 230; Page 9, line 266; Page 10, line 303)**.

Minor comments:

1. In Fig1f, the label of y-axis showed (%), which seems to be removed according to the text.

Response: Thanks for pointing out this. We apologize for not presenting this part correctly. We have corrected it in the **Revised Fig. 1f** and the legends in the **Revised manuscript (Page 27, lines 862-863)**.

2. Line 145-146, add a brief description of the phenotype of *Mettl3* cKO oocytes.

Response: Thanks for pointing out this. As "In *Mettl3*^{lox/lox}; *Gdf9-cre* (*Mettl3* conditional knockout (cKO)) mice, the size of the ovary largely decreases, and the oocytes are arrested from maturation. Few MII oocytes can be observed, and most of the oocytes are arrested at the GVBD stage. Moreover, the *Mettl3*^{Gdf9} cKO oocyte-derived zygotes are arrested at 1-cell stage after fertilization". We have added this description about the *Mettl3*^{Gdf9} cKO oocytes in the **Revised manuscript (Page 6, lines 151-155)**.

3. In sup Fig3g, the control group are the total RNAs, it's better to choose random RNAs with the same gene number as development-silencing RNAs.

Response: Thanks for pointing out this. We have randomly selected the same number (n=1478) of development-silencing RNAs as a control group to further evaluate the expression change. We still observed a significant difference between randomly selected RNAs and development-silencing RNAs, but not between the total RNAs and randomly chosen RNAs (**Rebuttal Fig. 3, Revised Supplementary Fig. 3g**).

Rebuttal Fig. 3. Oocyte development RNAs were more stable in *Mettl3*^{Gdf9} cKO oocytes.

Violin plot showing the expression change of total RNAs and the development-silencing RNAs between *Mettl3^{Gdf9}* cKO oocytes and control oocytes. The same number of the development-silencing RNAs was randomly selected from the total RNAs as non-sense control. *P* value was calculated by unpaired *t*-test.

4. The order of Fig4e and Fig4g seems to be wrong according to the text.

Response: Thanks for pointing out this. We apologize for this error and have corrected it in the **Revised manuscript (Page 30, lines 921-922, 925-927)** and **Revised Fig. 4**.

5. In Fig5b, how many genes are plotted? Number should be showed also in Sup Fig5g.

Response: Thanks for pointing out this. The detailed gene list and expression information have been included in **Original Table S7**. We have followed the reviewer's suggestion and added the numbers of modified and unmodified RNAs in each stage into the corresponding **Revised Fig. 5b-c** and **Revised Supplementary Fig. 5g**.

6. In Sup Fig6a, the variance among cells at the same stage seems too large to get the conclusion that the m⁶A level are decreasing after zygote.

Response: Thanks for the comments. We have modified this statement to “*We observed a trend of decreased m⁶A level after ZGA, which is consistent with the result detected by HPLC⁷*” in **Revised manuscripts (Page 10, lines 305-306)**. Moreover, the variance may be caused by the highly heterogeneous blastomeres of early embryos. Our result showed that the ZGA process is asynchronous in the 2-cell embryos, and the variance of m⁶A level decreases afterwards (**Original Fig. 6f,g**). Thus, we draw the conclusion of “m⁶A decreases after ZGA”.

7. The recent publication in Nature Cell Biology should be cited.

Response: Thanks for the kind reminding. Actually, we have already cited the publication of *Nature Cell Biology* in the Discussion section in the **Original manuscript**. Now, we have also added it in the Introduction section to introduce the low input m⁶A sequencing approach and in the Results section to evaluate the m⁶A characteristics of GV oocytes in the **Revised manuscript (Page 3, lines 75-76; Pages 5-6, lines 140-148)**.

Reviewer #2 (Remarks to the Author):

*In this study, Yao and colleagues develop a new technique of single-cell m⁶A sequencing (scm⁶A-seq) to profile the m⁶A methylome and transcriptome simultaneously. The authors observe the dynamic changes of m⁶A modified RNAs during oocyte maturation and early embryonic development. They find that m⁶A deficiency led to aberrant RNA clearance and low quality of oocytes in the model of *Mettl3* conditional knockout mouse. The asymmetric epi-transcriptome that is dependent on m⁶A further explains the heterogeneity between the blastomeres of 2-cell embryo. Overall, this study is carried out by the method of scm⁶A-seq in the single cell dimension to investigate in-depth into m⁶A characteristics and functions at the stage of gametogenesis and early embryonic development.*

Comments for the author

However, before publication, further interpretation and discussion of the data as well as additional experiments would be necessary to support the authors' conclusions.

Major comments:

1. Fig. 1D shows that the m⁶A modified fragments detected by scm⁶A-seq are the highest enrichment ratio in CDS region, which was not consistent with the previous reports that m⁶A was mainly found in 3'UTR, as mentioned in the Introduction (line 53). Please explain this difference.

Response: Thanks for pointing out this. To clearly explain the difference of m⁶A distribution, we first have searched the literatures about m⁶A distribution in human cancer cell lines, and found that this variation is probably derived from different methods. For m⁶A-SAC-seq, more than half of the sites are located in the 3' UTR region, and over 40% of m⁶A sites are located in the CDS region of polyA+ transcripts¹⁰. For DART-seq, there are nearly 70% of m⁶A sites in the 3' UTR region and about 20% in the CDS region¹¹. For m⁶A-REF-seq, almost 70% of detected m⁶A sites are located in CDS, and about 30% of the m⁶A sites are located in 3' UTR¹². Notably, for antibody-based methods, the distribution of detected m⁶A sites or peaks can be slightly influenced by different commercial sources of antibody¹³.

Another important factor affecting the distribution of m⁶A is the type of RNA for detection. There are much more m⁶A sites distributed in promoter, intron, and intergenic regions in total RNA samples than the polyA+ RNA samples¹⁴. Therefore, m⁶A can exist in any location of the transcriptome and is more enriched in long coding exons and 3' UTR of polyA+ RNAs. We have revised the description in the **Introduction** section in the **Revised manuscript (Page 3, lines 52-53)**.

Additionally, according to the previously reported methods^{8,15}, we have reanalyzed the data and included the stop codon region (stop codon ± 400 nt) when evaluating the distribution. Thus, we observed more than 40% m⁶A peaks located in 3'UTR and stop codon regions, and nearly 50% peaks in CDS region (**Rebuttal Fig. 4, Revised Fig. 1d**), which is consistent with the previous reports^{8,15}. Moreover, we had observed m⁶A enrichment around the stop codon region in scm⁶A-seq data (**Original Fig. 1e, Revised Fig. 1e**).

Overall, there is no significant difference in m⁶A distribution on mRNAs detected by scm⁶A-seq relative to other bulk m⁶A sequencing methods.

Rebuttal Fig. 4. The distribution of m⁶A peaks along mRNA in individual GV oocytes revealed by scm⁶A-seq. Barplot depicting the distribution of m⁶A peaks in different transcript segments.

2. As shown in Supplementary Fig. 1e, more than 10000 m6A modified RNAs are detected in the groups of 50 and 70 GV by bulk MeRIP-seq, while, the mounts of modified RNAs are about 5000 in the group of 25 GVs by the same way and so it is with single-cell through scm6A-seq. How to achieve the optimal balance between cell number and detection amount?

Response: Thanks for the thoughtful comment. To evaluate the balance between cell number and the detection amount for m⁶A, we have merged the single GV oocytes from scm⁶A-seq into gradient cell numbers for m⁶A detection. For each gradient, we randomly selected the same number of cells for five times and merged together. The results showed that the numbers of m⁶A modified peaks (**Rebuttal Fig. 5a**) and RNAs (**Rebuttal Fig. 5b**) increase along the increasing of cell number.

Considering the heterogeneity of single cells and the sensitivity of single cell sequencing¹, the superposition of several cells would indeed increase the number of identified m⁶A modifications. Moreover, the sequencing depth and genome coverage might also influence the m⁶A detection in the bulk sequencing data. Therefore, we have further analyzed the ULI-MeRIP-seq data of GV oocytes with three replicates⁷ to divide into gradient sequencing depth for m⁶A detection with the same method as above, and found that there was also a positive correlation between the sequencing depth and the detection amount of m⁶A (**Rebuttal Fig. 5c,d**).

For the results of bulk samples, it is more likely to reflect an average level of most cells, with the possibility of losing m⁶A characteristics for a small subtype of cells. Besides, the m⁶A modifications on highly expressed RNAs are more conserved and can easily be detected than those on under-expressed RNAs due to the calculation method¹³. We also obtained similar observations in this study (**Original Fig 1g,h, Revised Fig. 1g,h**).

Therefore, there is not a simple conclusion about the balance between the number of cells and the detection amount of m⁶A, which is related to the number of cells, genome coverage and analysis pipelines, etc. We have included this information in the discussion of the **Revised manuscript** (**Page 13, lines 383-387**).

Rebuttal Fig 5. The detected m⁶A number is positively correlated to sequencing depth.

a,b Box and whisker plot showing the identified m⁶A peaks (**a**) and m⁶A modified RNAs (**b**) in merged data sets of different cell numbers. **c,d** Box and whisker plot showing the identified m⁶A peaks (**c**) and m⁶A modified RNAs (**d**) using a part of the data of GV oocytes by ULI-MeRIP-seq⁷.

3. Some statements lack for sufficient evidence in the article, for example, Supplementary Fig. 1h suggests that scm⁶A-seq is more sensitive than MeRIP-seq through the expression and modification of *Hist4h4* gene; the integrated genomics viewer tracks of *Lars2* and *Mtus1* show the commonly and specifically m⁶A modified RNAs at different oocyte stages in Supplementary Fig. 4e, but it does not reflect the "far more" described in line 288. Please provide more evidence to support these descriptions.

Response: Thanks for these comments. To further illustrate the sensitivity of scm⁶A-seq for m⁶A detection, we have compared the data among scm⁶A-seq and bulk MeRIP-seq⁸ for GV oocytes and found that nearly 70% methylated RNAs in 2500 GVs by bulk MeRIP-seq were also identified in scm⁶A-seq (**Rebuttal Fig. 6a**). Moreover, we have performed the GO analysis and found that the conserved modified RNAs are annotated in cell cycle pathways, which is consistent with the previous study⁸ (**Rebuttal Fig. 6b**). In addition, there were 50% modified RNAs only detected in single GV by scm⁶A-seq, and these RNAs are related to DNA replication and cell cycle regulation pathways, such as *Hist1h4f* and *Mapk3* (**Rebuttal Fig. 6c, Revised Supplementary Fig. 1i**). To sum up, scm⁶A-seq is sensitive approach to detect both the highly conserved and low abundant m⁶A modifications which can most reflect the cellular heterogeneity (**Revised Fig. 1g,h**).

In addition, we apologize for lacking of the statement about stage-specific modification during oocyte maturation from GV to MII. We have corrected the description in the **Revised manuscript (Pages 8-9, lines 244-249)**, and reanalyzed and included the information about several more stage-specific modified RNAs (**Rebuttal Fig. 6d-g, Revised Supplementary Fig. 4g**).

Rebuttal Fig. 6. scm⁶A-seq sensitively reveals a dynamic m⁶A landscape during oocyte maturation

a. Venn diagram showing the commonly identified m⁶A modified RNAs in GV oocytes among scm⁶A-seq and bulk MeRIP-seq. **b.** The enriched Gene Ontology (GO) terms of 1450 commonly modified RNAs by different methods. **c.** Integrated genomics viewer (IGV) tracks showing the representative specific m⁶A peaks by scm⁶A-seq. **d-g** Integrated genomics viewer (IGV) tracks showing the stage-specific m⁶A peaks during oocyte maturation.

4. At present, some studies have detected m⁶A modified RNAs at the developmental stages of oocyte and pre-implantation embryos, such as references 6 and 10 cited in this paper. Has the scm⁶A-seq data gained from this study been compared with these publicly available data?

Response: Thanks for this comment. As discussed in comment #2, there are much fewer reads for an individual single cell generated from scm⁶A-seq libraries than bulk samples, which results in few m⁶A modified RNAs and m⁶A peaks being detected in an individual single cell (**Rebuttal Fig. 5**). Thus, we have used the aggraded scm⁶A-seq data of GV oocytes to address this thoughtful consideration. Because m⁶A sequencing data in reference 6 was generated from mouse embryonic stem cells instead of oocytes¹⁶, we didn't include it in this comparison analysis.

To compare the scm⁶A-seq data with previous publications, we have analyzed bulk MeRIP-seq data for GV oocytes with two replicates⁸, and low input ULI-MeRIP-seq data for GV oocytes with three replicates in another recent publication⁷, and compare them with the scm⁶A-seq data. We found that the modified RNAs detected by scm⁶A-seq cover nearly 70% of the modified RNAs from bulk MeRIP-seq and 55% of the modified RNAs from ULI-MeRIP-seq (**Rebuttal Fig. 7a, Revised Supplementary Fig. 1g**), and moreover, the 1746 conserved methylated RNAs were enriched in the critical biological processes for GV oocytes (**Rebuttal Fig. 7b, Revised Supplementary Fig. 1h**). Furthermore, we found that many vital mRNAs, such as *E2f1* and *Bod1*, are modified by m⁶A and are highly conserved among single cells (**Rebuttal Fig. 7c, Revised Supplementary Fig. 1i**).

Collectively, these results suggest the reliability and sensitivity of scm⁶A-seq.

Rebuttal Fig. 7. The m⁶A landscape in GV oocytes identified by different methods.

a. Venn diagram showing the overlap of identified m⁶A modified RNAs in GV oocytes by different methods. **b.** The enriched Gene Ontology (GO) terms of the commonly modified RNAs by different methods. **c.** Integrated genomics viewer (IGV) tracks showing the representative conserved m⁶A peaks in *E2f1* and *Bod1* mRNAs with detected by different methods. scm⁶A-seq data was also shown individually in single-cell resolution.

5. As observed in the study, m⁶A modification plays the opposite regulatory role at different stages, promoting mRNA degradation in GV oocytes while protecting the modified RNAs from degradation during MII-to-zygote transition. This phenomenon is interesting, please explain it further.

Response: Thanks for the wonderful comment. From our findings, the differential regulatory roles of m⁶A modification in mRNA metabolism might rely on various m⁶A-binding proteins in different categories of cells which mediate the methylated RNAs to different metabolic fates¹⁷. This phenomenon was also illustrated in the previous publication by *Kiaa1429* cKO GV oocytes and METTL3 inhibited cleavage embryos⁷.

We observed that *Ythdf2*, which mediates degradation of m⁶A modified RNAs, expresses at a relatively higher level in the growing oocytes than the fully-grown oocytes (**Revised Supplementary Fig. 3e**) based on the public data³, which supports by the previous publication⁴ about the YTHDF2 expression in the growing oocytes. Besides, we have observed a relative higher translation signal of *Ythdf2* in GV than MII oocytes and zygotes (**Rebuttal Fig. 8a, Revised Fig. 2e**) using public data⁵. Combined with the public eCLIP-seq data⁶, we also found that the METTL3-dependent m⁶A modified peaks are close to the YTHDF2 targets (**Revised Fig. 2f**) and more stable in *Mettl3^{Gdf9}* cKO oocytes (**Revised Fig. 2c, Revised Fig. 5d**). These results suggest that YTHDF2 is the probable regulator for m⁶A mediated RNA degradation in the GV oocytes.

In another case, m⁶A modified RNAs were more stable than the unmodified RNAs during MII-to-zygote stage (**Revised Fig 5b-d; Rebuttal Fig. 2; Revised Supplementary Fig. 5f**). Through analyzing the published data⁵, we have interestingly observed an obvious decrease in the translation signal of *Ythdf2* (**Rebuttal Fig. 8a, Revised Fig. 2e**), but a continued robust translation of *Igf2bp2* (**Rebuttal Fig. 8b, Revised Supplementary Fig. 5k**) from GV to MII stage, suggesting that IGF2BP2 may play a major role in stabilizing the methylated RNAs in MII oocytes. To explain the expression change of m⁶A modified RNAs during MII to zygote transition, we have sequenced the total transcriptome of both control and *Mettl3^{Gdf9}* cKO MII oocytes and zygote embryos. Interestingly, we found that the RNA metabolism is largely dysregulated from the *Mettl3^{Gdf9}* cKO oocytes to zygotes (**Rebuttal Fig. 8c, Revised Supplementary Fig. 5h**), and a higher proportion of m⁶A modified RNAs fall in the category of hypo-stable RNAs (**Rebuttal Fig. 8d, Revised Supplementary Fig. 5i**), suggesting that m⁶A contributes to the RNA instability in the *Mettl3^{Gdf9}* cKO oocytes (**Rebuttal Fig. 8e, Revised Fig. 5e**). These results were consistent with the finding that m⁶A increases the stability of m⁶A modified mRNAs in normal MII oocytes during MII-to-zygote transition.

Collectively, these results further illustrate that m⁶A promotes RNA degradation during oocyte growth and increases RNA stability during the MII-to-Zygote transition. More comprehensive experiments will be needed to further validate the precise role of m⁶A, such as the RNA substrate identification of reader proteins by eCLIP-seq and the metabolic analysis of m⁶A modified RNAs by RNA stability sequencing. We have included these discussions in the **Revised manuscript (Page 13, lines 388-400)**.

Rebuttal Fig. 8. Lacking m⁶A leads to aberrant MII-to-zygote transition.

a,b. Integrated genomics viewer (IGV) tracks showing the translation signal of *Ythdf2* (**a**) and *Igf2bp2* (**b**). Bigwig files RPKM normalized by 10 bp bins were downloaded from GEO (GSE165782). **c.** Scatterplot showing expression changes (zygote versus MII) between the *Mettl3*^{Gdf9} cKO group (n (MII) = 4, n (zygote) = 4) and control group (n (MII) = 4, n (zygote) = 5). |fold change| ≥ 4 was used as the cut-off to identify hyper-stable and hypo-stable genes. **d.** Bar plot showing the proportion of m⁶A modified and unmodified RNAs in hyper-stable and hypo-stable gene sets. **e.** Integrated genomics viewer (IGV) tracks showing the representative m⁶A-mediated stable RNAs during MII-to-zygote transition.

6. I would like the authors to reconsider the discussion. For example, some existing studies on RNA m⁶A modification in the field of reproduction should be included in the Introduction section in Introduction. In addition, most of discussion are merely repetition or summary of the results.

Response: Thanks for pointing out this. We have followed the reviewer's suggestion to include the studies on RNA m⁶A modification in reproduction field to the **Introduction** section in the **Revised manuscript**. Moreover, we have reorganized and further discussed the techniques and regulation of m⁶A during oocytes maturation and early embryo development in the **Discussion** section in the **Revised manuscript (Page 13, lines 388-401)**.

Minor comments:

1. The ordinate of Fig.1F is marked incorrectly.

Response: Thanks for pointing out this. We have corrected it in the legend of **Revised Fig. 1f**.

2. Line 157-158 has a syntax problem.

Response: Thanks for the helpful comments. We have corrected and modified the sentence in the **Revised manuscript (Page 6, lines 165-170)**.

3. Fig. 2e does not reflect the role of mettl3, although it is described in the legend, please describe it as consistently as Fig. 4e

Response: Thanks for this thoughtful suggestion. We apologize for the unclear description. We used the set of METTL3-dependent peaks to calculate the distance with YTHDF2 targets in

Original Fig. 2e. We have now clarified the legends clearly in **Revised Fig. 2f** and corrected the legends of **Revised Fig. 4e**.

4. Fig. 3g shows that the RNA abundance in NSN group is lower than that in the SN group. Which kind of GV oocyte was selected for scm⁶A-seq detection in the previous section?

Response: Thanks for this comment. We collected the GV oocytes from the ovary of four- to six-week-old female mice after ovulation induction using PMSG. Then, all the GV oocytes (>70 μm) were collected by puncturing the ovarian follicle and released with microcapillary pipettes in M2 media. We have clarified the procedures for the collection and culture of GV oocytes in the **Methods** section of the **Revised manuscript (Page 15, lines 443-447)**.

Usually, according to the developmental stage, the NSN oocytes refer to the growing oocytes with a smaller size, and the SN oocytes refer to the fully-grown oocytes with a larger size (>70 μm)¹⁸. SN GV oocytes have also been named as pSN oocytes in one recent study⁷. However, the NSN and SN oocytes could not simply be distinguished by their appearance with naked eyes, but by chromatin staining. Moreover, there are both SN and NSN oocytes for fully-grown GV oocytes identified according to the cell size¹⁹⁻²¹. Thus, we filter the fully-grown GV oocytes according to their size (> 70 μm) in our study without chromatin conformation staining. But we found scm⁶A-seq can distinguish SN and NSN oocytes by combining the transcriptome and m⁶A methylome (**Original Fig. 3**). The difference between NSN oocytes and SN oocytes is one of the characteristics between these two kinds of fully-grown oocytes because the NSN oocytes are still in transcription activation with uncompleted process of RNA accumulation, just as the SN oocytes do.

5. Is the order of Fig. 4e and Fig. 4G reversed?

Response: Thanks for pointing out this and we apologize for this error. We have corrected it in the **Revised manuscript (Page 30, lines 921-922 and lines 925-927)**, and checked all the figures to ensure the accuracy of all the information.

References

1. Tegowski, M., Flamand, M.N. & Meyer, K.D. scDART-seq reveals distinct m⁶A signatures and mRNA methylation heterogeneity in single cells. *Mol Cell* **82**, 868-878 (2022).
2. Hu, Y., *et al.* Oocyte competence is maintained by m⁶A methyltransferase KIAA1429-mediated RNA metabolism during mouse follicular development. *Cell Death Differ* **27**, 2468-2483 (2020).
3. Xu, Q.H., *et al.* SETD2 regulates the maternal epigenome, genomic imprinting and embryonic development. *Nat Genet* **51**, 844-856 (2019).
4. Ivanova, I., *et al.* The RNA m⁶A reader YTHDF2 is essential for the post-transcriptional regulation of the maternal transcriptome and oocyte competence. *Mol Cell* **67**, 1059-1067 (2017).
5. Xiong, Z., *et al.* Ultrasensitive Ribo-seq reveals translational landscapes during mammalian oocyte-to-embryo transition and pre-implantation development. *Nat Cell Biol* **24**, 968-980 (2022).
6. Lasman, L., *et al.* Context-dependent functional compensation between Ythdf m⁶A reader proteins. *Genes Dev* **34**, 1373-1391 (2020).
7. Wu, Y., *et al.* N⁶-methyladenosine regulates maternal RNA maintenance in oocytes and timely RNA decay during mouse maternal-to-zygotic transition. *Nat Cell Biol* **24**, 917-927 (2022).
8. Mu, H.Y., *et al.* METTL3-mediated mRNA N⁶-methyladenosine is required for oocyte and follicle development in mice. *Cell Death Dis* **12**, 989 (2021).
9. Mendel, M., *et al.* Methylation of structured RNA by the m⁶A writer METTL16 is essential for mouse embryonic development. *Mol Cell* **71**, 986-1000 e1011 (2018).
10. Hu, L.L., *et al.* m⁶A RNA modifications are measured at single-base resolution across the mammalian transcriptome. *Nat Biotechnol* **40**, 1210-1219 (2022).
11. Meyer, K.D. DART-seq: an antibody-free method for global m⁶A detection. *Nat Methods* **16**, 1275-1280 (2019).
12. Zhang, Z., *et al.* Single-base mapping of m⁶A by an antibody-independent method. *Sci Adv* **5**, eaax0250 (2019).
13. Zeng, Y., *et al.* Refined RIP-seq protocol for epitranscriptome analysis with low input materials. *PLoS Biol* **16**, e2006092 (2018).
14. Linder, B., *et al.* Single-nucleotide-resolution mapping of m⁶A and m⁶Am throughout the transcriptome. *Nat Methods* **12**, 767-772 (2015).
15. Dominissini, D., *et al.* Topology of the human and mouse m⁶A RNA methylomes revealed by m⁶A-seq. *Nature* **485**, 201-U284 (2012).
16. Wei, J.B., *et al.* FTO mediates LINE1 m⁶A demethylation and chromatin regulation in mESCs and mouse development. *Science* **376**, 968-973 (2022).
17. Zhao, Y., Shi, Y., Shen, H. & Xie, W. m⁶A-binding proteins: the emerging crucial performers in epigenetics. *J Hematol Oncol* **13**, 35 (2020).
18. Ma, J.Y., *et al.* Maternal factors required for oocyte developmental competence in mice: transcriptome analysis of non-surrounded nucleolus (NSN) and surrounded nucleolus (SN) oocytes. *Cell Cycle* **12**, 1928-1938 (2013).
19. Cheng, S., *et al.* Mammalian oocytes store mRNAs in a mitochondria-associated membraneless compartment. *Science* **378**, eabq4835 (2022).
20. Wang, T. & Na, J. Fibrillarin-GFP facilitates the identification of meiotic competent oocytes. *Front Cell Dev Biol* **9**, 648331 (2021).
21. Zuccotti, M., Piccinelli, A., Giorgi Rossi, P., Garagna, S. & Redi, C.A. Chromatin organization during mouse oocyte growth. *Mol Reprod Dev* **41**, 479-485 (1995).

REVIEWERS' COMMENTS

Reviewer #1 (Remarks to the Author):

The authors have conducted additional experiments and analysis, and successfully addressed all the concerns raised by this reviewer.

Reviewer #2 (Remarks to the Author):

I thank the authors for all their hard work in addressing the previous concerns and points raised by me and the other reviewers. I am satisfied with the extra work that has been done and find this manuscript has been improved. Although, I fully support publication and have no other major comments, two minor scientific comments need to be fixed before publication:

1. There are some cases in the revised manuscript where the description do not correspond to the figures. For example, Supplementary Fig. 4f appears incorrectly on line 249. The description in line 252-254 corresponds to Supplement Fig. 5i rather than Supplement Fig. 5h. Supplement Fig. 6c is not referenced in the manuscript. There are other errors in the text. Please check them carefully. Please correct these errors and examine the manuscript carefully.
2. The sample distribution in Fig. 3c and Fig. 3e is consistent, but some loci in the CKO group in Fig. 3c are transformed into the control group in Fig. 3e, and some loci in the control group in Fig. 3c belong to the CKO group in Fig. 3e. Please explain this phenomenon of conversion.

2. Point-by-point responses to reviewers

REVIEWERS' COMMENTS

Reviewer #1 (Remarks to the Author):

The authors have conducted additional experiments and analysis, and successfully addressed all the concerns raised by this reviewer.

Response: We thank the reviewer for his/her positive recognition of our work, and are very grateful to the insightful comments and suggestions.

Reviewer #2 (Remarks to the Author):

I thank the authors for all their hard work in addressing the previous concerns and points raised by me and the other reviewers. I am satisfied with the extra work that has been done and find this manuscript has been improved. Although, I fully support publication and have no other major comments, two minor scientific comments need to be fixed before publication:

Response: We are grateful to this reviewer's positive comments, and have followed the reviewer's comments to revise our manuscript accordingly.

1. There are some cases in the revised manuscript where the description do not correspond to the figures. For example, Supplementary Fig. 4f appears incorrectly on line 249. The description in line 252-254 corresponds to Supplement Fig. 5i rather than Supplement Fig. 5h. Supplement Fig. 6c is not referenced in the manuscript. There are other errors in the text. Please check them carefully. Please correct these errors and examine the manuscript carefully.

Response: We apologize for our negligence and appreciate the reviewer for pointing out these errors. We have corrected these errors in the **Revised Manuscript**, and carefully checked our manuscript to ensure the results being described clearly and correctly (**Pages 9, line 255; Pages 9-10, lines 273-283; Page 10, lines 291-294; Pages 10-11, lines 312-314**).